# RLAD: Training LLMs to Discover Abstractions for Solving Reasoning Problems

**Yuxiao Qu**[1,*], **Anikait Singh**[2,*], **Yoonho Lee**[2,*], **Amrith Setlur**[1]
**Ruslan Salakhutdinov**[1], **Chelsea Finn**[2] **Aviral Kumar**[1]

[1]Carnegie Mellon University [2]Stanford University *Equal Contribution

## Abstract

Reasoning requires going beyond pattern matching or memorization of solutions to identify and implement "algorithmic procedures" that can be used to deduce answers to hard problems. Doing so requires reusing primitives, intermediate results, or procedures across multiple problems. While RL post-training on long chains of thought ultimately aims to uncover this kind of algorithmic behavior, the depth-first and "brute-force" nature of reasoning traces learned by these models suggests that this is far from a fulfilled promise. To address more effective reasoning, we introduce *reasoning abstractions*: concise natural language descriptions of procedural and factual knowledge that guide the model toward learning successful reasoning. We train models to be capable of proposing several useful abstractions given a problem, followed by RL training that incentivizes building a solution while using the information provided by these abstractions. This results in a two-player RL training paradigm, abbreviated as RLAD, that jointly trains an abstraction generator and an abstraction-conditioned solution generator. This setup effectively enables structured exploration, decouples learning signals of abstraction proposal and solution generation, and improves generalization to harder problems. We also show that spending more test-time compute into generating abstractions is more beneficial for performance than generating more solutions at large inference-time budgets, illustrating the role of abstractions in guiding global exploration.

## 1 Introduction

Modern machinery for training large language models (LLMs) to reason relies on incentivizing longer chains of thought via reinforcement learning (RL). This training approach largely incentivizes "depth": subsequent training iterations increase response length by incorporating new operations that usually verify or build on top of the line of reasoning being already pursued by the model (Setlur et al., 2025). This often results in very long chains of thought that appear to explore the solution search space, but in a sequential, brute-force manner. In many problems, it is instead more desirable to optimize for "breadth": explore a diverse array of solution strategies, rather than committing to a seemingly good set of reasoning strategies right away (Yu et al., 2025; Yue et al., 2025). Even though models trained this way succeed on some problems, they fail on problems with similar difficulty, revealing poor generalization (Ma et al., 2024; Mirzadeh et al., 2024; Petrov et al., 2025).

***How can we help models explore a breadth of reasoning strategies for a given problem?*** Abstractly, the most natural approach is to train models to hypothesize new strategies to attack difficult problems and then attempt to utilize these strategies in the solution. We can do this by making models capable of discovering ***reasoning abstractions***: compressed representations of shared procedures that underlie multiple candidate solutions to a problem. For example, in math reasoning, such abstractions might correspond to useful intermediate lemmas or even some intermediate steps that do not succeed but illustrate what not to do. When presented in context, these abstractions function akin to "hints" on an exam, enabling LLMs to solve harder problems by building on the insights appearing in the abstraction. That is, when conditioned on abstractions, training via RL should train the LLM to implement useful meta strategies that utilize and compose the procedural information in the abstraction as best as possible to solve the problem, rather than attempting to search over the procedural information itself. This naturally boosts the diversity of solution strategies and behaviors that a model learns to utilize when encountering an unseen problem, in contrast to committing to a

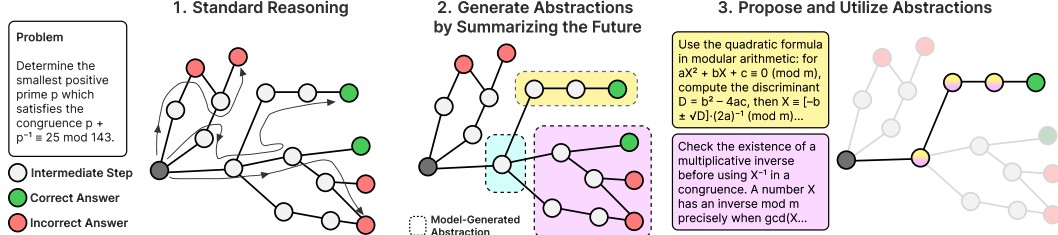

Figure 1: *Reasoning abstractions illustrated in the solution-space graph for a problem.* We depict the solution space as a graph of intermediate steps leading to correct or incorrect answers. (1) Standard reasoning explores this space along one sequential chain. (2) We generate textual abstractions by summarizing which intermediate steps led to which outcomes. (3) Such abstractions can be reused to guide reasoning more efficiently.

narrow set of approaches. In RL terminology, abstractions serve as high-level subgoals, skills, or priors—any of them depending upon context—guiding the low-level solution-generating policy.

In this work, we imbue LLMs with the capability of proposing and utilizing abstractions for solving problems. Concretely, we build reasoning models that, first, given an input problem, propose one or more reasoning abstractions, expressed in natural language. Subsequently, they generate a solution that utilizes the information and principles prescribed by these abstractions. To achieve this, we jointly train two LLMs via RL post-training: **(1)** an abstraction generator, and **(2)** an abstraction-conditioned solution generator. The abstraction generator is rewarded for the improvement in the accuracy of the solution generator, stemming from conditioning on the abstractions it proposes. The solution generator is rewarded to maximize accuracy in solving a problem when using the abstraction. To obtain a good initialization for RL training, we warmstart both models by running supervised fine-tuning (SFT) on data from stronger models. For the abstraction generator, we collect multiple candidate solutions and prompt a stronger LLM to generate diverse abstractions. For the solution generator, we generate solutions conditioned on an abstraction.

The main contribution of this paper is the notion of *reasoning abstractions*, how they can be obtained, training procedures to amplify them via RL training, and an illustration of how they can be used to improve reasoning performance and exploration of the search space. Concretely, we build an approach to imbue LLMs with the capability of proposing abstractions, and evaluate the model on a variety of math-reasoning benchmarks, AIME 2025 (Mathematical Association of America, 2025), DeepScaleR Hard (Setlur et al., 2025), and AMC 2023. We find an average 44% improvement over state-of-the-art long chain-of-thought RL approaches (i.e., DAPO (Yu et al., 2025)) on AIME 2025, and show an effective benefit from generating diverse abstractions over brute-force solution sampling.

## 2 RELATED WORK

**Scaling test-time compute and exploration.** Recent work highlights the promise of scaling test-time compute in different ways. One approach involves parallel sampling: sampling multiple reasoning rollouts and then selecting a winner via a scoring rule (Uesato et al., 2022; Wang et al., 2023; Charniak & Johnson, 2005; Feng et al., 2024; Snell et al., 2024; Yao et al., 2023a; Hao et al., 2023; Snell et al., 2024). A complementary line of work iteratively edits a single trace, attempting to implement some sort of a sequential search within a single solution trace (Madaan et al., 2023; Qu et al., 2024; Qu et al., 2024; Kumar et al., 2024). As such, the sequential approach performs a bit worse on harder problems (Snell et al., 2024; Qu et al., 2025), where it often gets trapped in strategies that seem optimal but aren't actually (Pan et al., 2025). Yet it still performs better than parallel search on easier and medium difficulty problems (Snell et al., 2024). Our approach of proposing and leveraging abstractions enables a kind of a hybrid between sequential sampling and parallel sampling, guided by the proposed abstractions. Some concurrent work (Pan et al., 2025) studies directly interleaving parallel and sequential samples, and while it is similar in theory to us, it only distills this interleaved structure into the model and does not run RL training to optimize parallel and sequential sampling procedures employed here. Prior work has also utilized hand-designed scaffolds to integrate multi-step evaluations of intermediate hypotheses into reasoning (Yao et al., 2023b; Ho et al., 2023; Hao et al., 2023; Li et al., 2023). In contrast, we do not rely on pre-defined interfaces but learn to *automatically* propose useful abstractions.

**Using prior knowledge for LLM reasoning.** Several threads of work converge on the idea that *textual artifacts* such as examples, plans, or prompts, can serve as reusable knowledge that steers

LLM behavior. Existing retrieval-augmented generation (RAG) pipelines assume a static corpus, typically of human-written text, and focus on improving retrieval heuristics (Lewis et al., 2020; Borgeaud et al., 2022; Trivedi et al., 2022; Verma et al., 2024; Anonymous, 2025; Li et al., 2025). Many works use LLMs to learn or refine prompts, either in an input-agnostic fashion (Zhou et al., 2022; Yang et al., 2023; Pryzant et al., 2023; Fernando et al., 2023) or through input-specific edits based on feedback (Shinn et al., 2023; Madaan et al., 2023; Gou et al., 2023; Yuksekgonul et al., 2025; Lin et al., 2025). Other related work explores the use of synthetic demonstrations (Zelikman et al., 2022b), scratchpads (Nye et al., 2021), and memory-augmented agents (Schäfer et al., 2020) to encode prior problem-solving knowledge. Two recent works demonstrate that LLMs can accumulate and reuse their own experience across tasks (Zhao et al., 2024; Suzgun et al., 2025). While one can view our abstractions as a form of prior procedural and factual knowledge produced before the model's solution attempt, this knowledge is **(a)** input-dependent and **(c)** is not acquired from an external source at deployment, but rather is "proposed" by the model itself. Imbuing models with this capability requires a two-player RL training process. To our knowledge, such procedures have not been used for generating textual artifacts of any type, let alone the abstractions we consider.

## 3 PRELIMINARIES AND NOTATION

We study reasoning with LLMs, where the LLM is provided access to a problem $\mathbf{x}$, and generates a stream of tokens $\mathbf{y}$ that ends in an estimate of the answer. We assume access to a rule-based ground-truth 0/1 reward $\mathrm{Acc}_{\mathbf{x}}(\mathbf{y}, \mathbf{y}^\star) \in \{0, 1\}$ that measures correctness of the produced answer $\mathbf{y}$, against the ground-truth solution $\mathbf{y}^\star$ for a question $\mathbf{x}$. For training, we are given a dataset $\mathcal{D}_{\mathrm{train}} = \{(\mathbf{x}_i, \mathbf{y}_i^\star)\}_{i=1}^N$ of problems $\mathbf{x}_i$ and solutions $\mathbf{y}_i^\star$ that end with the correct answer. Our goal is to train the LLM $\pi(\cdot|\mathbf{x})$ such that it achieves high rewards on a test distribution of problems $\mathcal{P}_{\mathrm{test}}$.

We evaluate models via average accuracy under $\mathcal{P}_{\mathrm{test}}$. We also measure the pass@k metric, where for problem $\mathbf{x}$, we sample $k$ solutions $\mathbf{y}_1, \ldots, \mathbf{y}_k \sim \pi(\cdot|\mathbf{x})$, and consider the problem to be solved if any of these $k$ traces is correct. This metric couples accuracy with diversity, i.e., it attains the largest value when the model effectively finds diverse, good responses. To reduce variance in estimating pass@k, we sample $n \geq k$ samples per problem and use the unbiased estimator introduced in OpenAI Codex (Chen et al., 2021): $1 - \binom{n-c}{k}/\binom{n}{k}$, where $c \leq n$ is the number of correct samples.

## 4 REASONING ABSTRACTIONS AND WHY THEY ARE USEFUL

Solving reasoning problems often requires composing both *procedural* knowledge (e.g., how to apply a root-finding algorithm) and *factual* knowledge (e.g., relevant lemmas or intermediate results). Current approaches train models to reason via reinforcement learning (RL) on long chains of thought. However, this is often ineffective as RL often tends to optimize for "depth", producing longer traces where each subsequent segment builds on the last segment (e.g., verifying prior calculations), rather than "breadth", i.e., exploring diverse solution strategies or utilizing seemingly irrelevant procedures when needed. We now introduce the concept of ***reasoning abstractions***, concise insights that explicitly encode a range of useful procedural and factual concepts for a problem. We describe our approach for generating abstractions and demonstrate mechanisms that make them work.

### 4.1 PROPOSING GOOD REASONING ABSTRACTIONS BY SUMMARIZING SOLUTION ATTEMPTS

To imbue LLMs with the capability of proposing abstractions, we warmstart the LLM using a dataset consisting of problems paired with a small set of high-quality reasoning abstractions, created synthetically. Perhaps the most natural way to obtain an initial set of reasoning abstractions is to collect a diverse set of traces attempting to solve a problem and then summarize useful concepts appearing in these traces (see Figure 1 for an illustration). More formally, consider the space of possible reasoning traces for a given problem as a graph where the nodes of the graph represent intermediate states encountered when solving a question (see Figure 1). Good abstractions would identify useful substructures within this larger reasoning graph. For example, an abstraction can capture whether a set of strategies lead to a similar outcome or another set of tactics leads to an error being consistently made.

**Generating abstractions.** We now quantitatively evaluate whether we can generate useful abstractions by summarizing key insights in reasoning and solution traces that solve a problem. To do so, we prompt a model to generate solution traces and prompt a stronger model to deduce useful patterns from the responses of the first model. Concretely, we utilize the Qwen3 (Qwen Team, 2025) series of models to produce solutions and a stronger reasoning model, o4-mini, to generate abstractions.

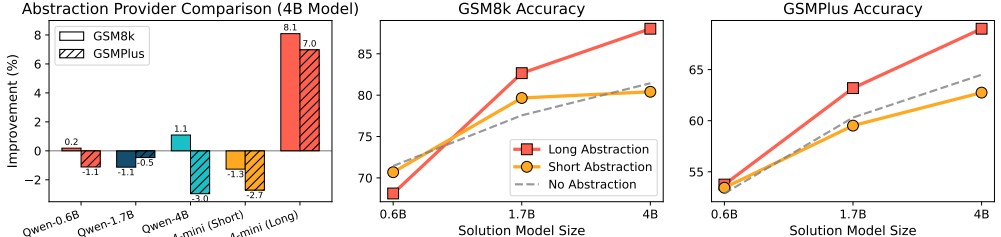

Figure 2: *Benefits from abstractions hinge on solver scale, abstraction length, and solution model*. Most configurations fail to yield gains; only o4-mini with long and detailed abstractions shows consistent improvements across the GSM8k and GSMPlus datasets (left). Solver capability also matters: even strong abstractions help only if the solution model is sufficiently capable (middle, right).

While this approach is not perfect, it enables us to validate the feasibility of the concept of reasoning abstractions and generate warmstart data for *training* LLMs to propose abstractions. To ensure that the abstractions do not "leak" content of the solution, we verify post-hoc that prompting a model with only the abstraction and no problem yields zero accuracy when sampling 16 times from the base model. This makes these abstractions suitable for our study as they only provide useful information while not allowing the model to shortcut to the answer.

**Evaluating abstractions.** To validate the efficacy of using abstractions, we adopt a simple test based on performance after conditional generation. Concretely, let us denote the LLM policy that produces a solution conditioned on the problem $\mathbf{x}$ as $\pi_\theta^{\mathrm{sol}}(\cdot|\mathbf{x})$. A good abstraction $\mathbf{z}$ is a sequence of tokens that provides some useful procedural and factual information to improve model performance:

$$\mathbb{E}_{\widetilde{\mathbf{y}}\sim\pi_\theta^{\mathrm{sol}}(\cdot|\mathbf{x},\mathbf{z})}\left[\mathrm{Acc}(\widetilde{\mathbf{y}},\mathbf{y}^*)\right] > \mathbb{E}_{\widetilde{\mathbf{y}}\sim\pi_\theta^{\mathrm{sol}}(\cdot|\mathbf{x})}\left[\mathrm{Acc}(\widetilde{\mathbf{y}},\mathbf{y}^*)\right]. \qquad (1)$$

## 4.2 RESULTS AND OBSERVATIONS

**Evaluation on math reasoning.** After generating abstractions, we measure their quality by evaluating Equation 1, i.e., by checking if conditioning the problem solver on a set of abstractions improves its accuracy. Results in Figure 2 show that conditioning a problem solver on abstractions improves accuracy when two conditions hold simultaneously: (i) the abstraction is not too short (e.g., not just a few words that are not informative; example in Appendix D.2) and is generated by a strong generator (o4-mini) and (ii) the solution generator has sufficient instruction-following capability (Qwen3-1.7B or Qwen3-4B) of interpreting and utilizing the generated abstraction. These results confirm that good abstractions (satisfying equation 1) exist for math problems, but neither the ability to generate them nor the ability to leverage them in solutions arises naturally. In Section 5, we will describe our method for explicitly training models to propose and use such abstractions effectively.

**Evaluation on ARC-AGI.** We also evaluate abstractions on the ARC-AGI benchmark. We present some details of our setup in Appendix B.3. We evaluate on 90 ARC puzzles evenly derived from the test sets of ARC-AGI 1, ARC-AGI 2, and BARC (Li et al., 2024). In Table 1, we present the pass@k and coverage performance (% of unit tests successfully passed) of the base Qwen3-4B model when conditioned on a proposed abstraction vs not using any abstraction. We see a positive improvement in both metrics on this domain over multiple samples, indicating an improvement from utilizing reasoning abstractions.

| k | pass@k | | coverage | |
|---|---|---|---|---|
| | w/o abs | w/ abs | w/o abs | w/ abs |
| 1 | 14.0% | **18.0%** | 19.5% | **22.5%** |
| 2 | 17.5% | **22.5%** | 24.0% | **28.5%** |
| 4 | 20.0% | **26.5%** | 28.5% | **34.0%** |
| 8 | 22.8% | **30.2%** | 31.8% | **38.8%** |
| 16 | 24.7% | **33.2%** | 35.0% | **42.9%** |

Table 1: *pass@k accuracy and max@k coverage on ARC-AGI*. Abstractions yield consistent gains in both metrics.

**Interpreting the generated abstractions.** We also show some examples of the discovered abstractions in Appendix D.3. We now attempt to interpret these abstractions by classifying them into several categories. We observe that the discovered abstractions often correspond to useful techniques (e.g., "launchpoint" in Appendix D.3), a useful lemma or heuristic principle (e.g., "blind-follow" in Appendix D.3), and cautionary examples that demonstrate common pitfalls encountered when solving a problem (e.g., "caution alert" in Appendix D.3). These abstractions distill complex reasoning patterns and potential approaches into useful nuggets, allowing models to generalize across structurally similar problems. Finally, we want to remind the reader and emphasize that these qualitative results from interpreting the discovered abstractions are specific to an individual problem, and not representative of the process being used to discover them. Our approach for generating abstractions is neither hand-engineered for interpretability or uses any such heuristics beyond summarization.

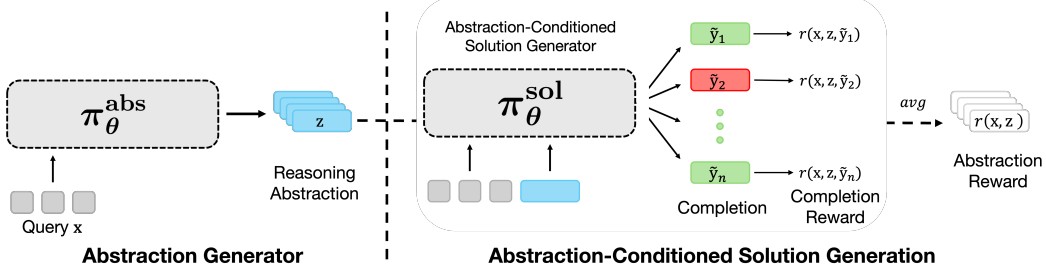

Figure 3: **RLAD *training paradigm.*** We train an abstraction generator, $\pi_\theta^{\mathrm{abs}}$, that proposes some reasoning abstractions conditioned on the question $\mathbf{x}$, denoted as $\mathbf{z}$. Then, the solution generator, $\pi_\theta^{\mathrm{sol}}$, is trained to produce a response, $\tilde{\mathbf{y}}$, conditioned on the generated abstraction $\mathbf{z}$. The reward used for training $\pi_\theta^{\mathrm{abs}}$ corresponds to the average success rate of the solution generator conditioned on the proposed abstraction.

**Good abstractions exist in many domains.** We also find that the summarization procedure can be used to identify an initial set of useful reasoning abstractions on many problem domains, including healthcare, human behavior, legal reasoning, and web security. Of course, the proportion of an abstraction devoted to procedural knowledge and factual knowledge is different in these domains compared to math. That said, we find that using reasoning abstractions improves performance by 30% on average over 37 tasks from RAFT (Alex et al., 2021), CLUES (Menon et al., 2022), and LegalBench (Guha et al., 2023). We show some examples in Figure 7 and full results in Appendix B.3.

> **Takeaways: Reasoning abstractions summarize insights useful for guiding solution traces**
>
> Reasoning abstractions summarize procedural and factual knowledge that is useful for learning to solve problems via diverse strategies. Proposing abstractions generated by summarizing solution traces already improves performance of base generators by 30% for math reasoning.

## 5 RLAD: LEARNING TO PROPOSE REASONING ABSTRACTIONS

Having defined the notion of reasoning abstractions and shown that they can improve performance when adhered to for reasoning, we will now develop an approach to train LLMs to be capable of both proposing and utilizing abstractions. Doing so requires training an *abstraction generator*: an LLM, $\mathbf{z} \sim \pi_\theta^{\mathrm{abs}}(\cdot|\mathbf{x})$ that proposes candidate abstractions $\mathbf{z}$ given problem $\mathbf{x}$, and an abstraction-conditioned solution generator, $\mathbf{y} \sim \pi_\theta^{\mathrm{sol}}(\cdot|\mathbf{x}, \mathbf{z})$, that produces a solution $\mathbf{y}$ given $\mathbf{x}$ and abstraction $\mathbf{z}$. Note that $\mathbf{z}$ is parameterized as a variable-length sequence of tokens and might consist of one or more facts or procedures. While our approach applies to the case when $\pi_\theta^{\mathrm{abs}}$ produces more than one abstraction, we abuse notation and subsume multiple abstraction into one to avoid clutter. We now describe RL with abstraction discovery (RLAD), our method for training these models via RL.

### 5.1 TRAINING $\pi_\theta^{\mathrm{abs}}$ AND $\pi_\theta^{\mathrm{sol}}$ VIA RL

The core principle behind our approach is that an abstraction $\mathbf{z}$ is successful at a given problem $\mathbf{x}$ if it can maximally help $\pi_\theta^{\mathrm{sol}}(\cdot|\mathbf{x}, \mathbf{z})$ find correct responses to the question $\mathbf{x}$, without actually leaking the answer itself. To convert this into an RL objective, we design a reward function that rewards an abstraction $\mathbf{z}$ with the expected success of solutions generated by $\pi_\theta^{\mathrm{sol}}$ conditioned on $\mathbf{z}$:

$$r_{\pi_\theta^{\mathrm{sol}}}(\mathbf{x}, \mathbf{z}) := \mathbb{E}_{\widetilde{\mathbf{y}} \sim \pi_\theta^{\mathrm{sol}}(\cdot|\mathbf{x}, \mathbf{z})}\left[\mathrm{Acc}_{\mathbf{x}}(\widetilde{\mathbf{y}}, \mathbf{y}^*)\right], \tag{2}$$

where $\mathbf{y}^*$ is the ground-truth answer and $\mathrm{Acc}_{\mathbf{x}}(\cdot, \cdot)$ denotes the 0/1 accuracy on problem $\mathbf{x}$. To train $\pi_\theta^{\mathrm{sol}}$, one can then adopt the fairly straightforward approach of maximizing 0/1 binary outcome reward, now conditioned on a given abstraction $\mathbf{z}$ sampled previously from $\pi_\theta^{\mathrm{abs}}$, akin to recent results RL (DeepSeek-AI et al., 2025). Formally, we set the reward for a solution as: $r(\mathbf{x}, \mathbf{z}, \widetilde{\mathbf{y}}) := \mathrm{Acc}_{\mathbf{x}}(\widetilde{\mathbf{y}}, \mathbf{y}^*)$. With these reward functions in place, perhaps the most natural approach then would be to train $\pi_\theta^{\mathrm{abs}}$ to maximize $r_{\pi_\theta^{\mathrm{sol}}}$ for a fixed $\pi_\theta^{\mathrm{sol}}$ on a dataset of prompts $\mathcal{D}_{\pi_\theta^{\mathrm{abs}}}$, while also iteratively training $\pi_\theta^{\mathrm{sol}}$ to maximize the reward function $r$ on modified prompts generated by concatenating a set of sampled abstraction $\mathbf{z}$ on a dataset of problems, $\mathcal{D}_{\pi_\theta^{\mathrm{sol}}}$. This maximization could be done via on-policy RL (e.g., GRPO (Shao et al., 2024)) or offline RL methods (e.g., DPO (Rafailov et al., 2023), STaR (Zelikman et al., 2022a)). This represents a co-operative two-player game.

**Challenges with naïve reward design.** While the approach so far is extremely simple, it presents some challenges. In particular, the reward functions defined above can result in undesirable solutions

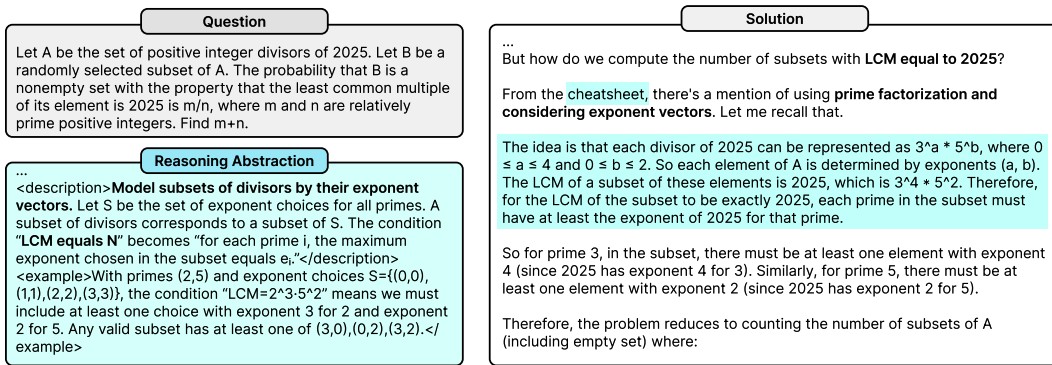

Figure 4: *A typical example of a reasoning abstraction proposed by our abstraction generator.* In the solution, we see (**in blue**) references to the abstraction ("cheatsheet") and keywords from the abstraction being used meaningfully in the reasoning trace of the solution generator model.

in a rather nuanced manner: **(1)** if $\pi_\theta^{\mathrm{abs}}$ learns to solve problem $\mathbf{x}$ in its entirety, it will still be rewarded highly by $r_{\pi_\theta^{\mathrm{sol}}}$ but is not a desirable abstraction; **(2)** if $\pi_\theta^{\mathrm{sol}}$ is too weak or too strong, such that it is either able to always solve the problem $\mathbf{x}$ or never solves it, then $r_{\pi_\theta^{\mathrm{sol}}}$ will not provide a meaningful signal to update $\pi_\theta^{\mathrm{abs}}$; and **(3)** similar to the above failure modes, training $\pi_\theta^{\mathrm{sol}}$ via on-policy RL may result in it ignoring the abstraction $\mathbf{z}$ altogether no matter how useful it is. Abstractly, all of these challenges stem from an asymmetry in the strength of $\pi_\theta^{\mathrm{abs}}$ and $\pi_\theta^{\mathrm{sol}}$, where one may drown out the learning signal for the other. We therefore build a modified reward system for training.

**Modifying reward design.** We make a small but consequential change to the training reward system. In particular, we train $\pi_\theta^{\mathrm{sol}}$ on a mixture of prompts $\mathbf{x}$ augmented by abstractions $\mathbf{z}$ and prompts $\mathbf{x}$ without any abstractions at all. In this process, while we utilize $\mathrm{Acc}_\mathbf{x}$ as discussed above on a given response, we simply zero out rewards for any trace generated on $\mathbf{x}$ without abstractions. When utilizing KL-constrained RL, e.g., GRPO (Shao et al., 2024), $\pi_\theta^{\mathrm{sol}}$ is now trained to closely mimic the distribution of responses as the reference LLM on questions $\mathbf{x}$ but must attempt to find ways to optimize reward on the same question $\mathbf{x}$ when augmented with an abstraction. This can be accomplished only when $\pi_\theta^{\mathrm{sol}}$ learns to utilize the provided abstraction carefully, hence addressing one of the challenges above. Formally, the updated versions of these reward functions are shown as:

$$r(\mathbf{x}, \mathbf{z}, \widetilde{\mathbf{y}}) := \begin{cases} 0, & \text{if } \mathbf{z} = \emptyset \\ \mathrm{Acc}_\mathbf{x}(\widetilde{\mathbf{y}}, \mathbf{y}^*), & \text{otherwise} \end{cases} \tag{3}$$

$$r_{\pi_\theta^{\mathrm{sol}}}(\mathbf{x}, \mathbf{z}) := \mathbb{E}_{\widetilde{\mathbf{y}} \sim \pi_\theta^{\mathrm{sol}}(\cdot | \mathbf{x}, \mathbf{z})}[\mathrm{Acc}_\mathbf{x}(\widetilde{\mathbf{y}}, \mathbf{y}^*)]. \tag{4}$$

## 5.2 Warmstarting $\pi_\theta^{\mathrm{sol}}$ and $\pi_\theta^{\mathrm{abs}}$ from Good Initializations

While the above approach prescribes a recipe for RL training of $\pi_\theta^{\mathrm{abs}}$ and $\pi_\theta^{\mathrm{sol}}$, any such recipe critically relies on the ability of the initialized model to generate somewhat meaningful abstractions and meaningful solutions conditioned on the abstraction input, respectively, right from the beginning of RL training. Inspired from the approach of running an initial phase of SFT to imbue into the model the basic structure of a long chain-of-thought before RL (DeepSeek-AI et al., 2025; Qu et al., 2025), we run an initial phase of SFT to imbue into $\pi_\theta^{\mathrm{abs}}$ and $\pi_\theta^{\mathrm{sol}}$ the basic capabilities of producing abstractions and attempting to follow abstractions respectively, even if the resulting models are not very good. For this initial warmstart phase, we follow the protocol from Section 4 and construct a corpus $\{(\mathbf{x}_i, \mathbf{z}_i, \mathbf{y}_i)\}_{i=1}^M$ by prompting strong models. For each training problem-solution pair $(\mathbf{x}, \mathbf{y}^*)$, in our training set, we first generate an abstraction $\mathbf{z}$ using an instruction-tuned model, discarding any that leak $\mathbf{y}^*$. We then sample a solution trace $\mathbf{y}$ conditioned on $(\mathbf{x}, \mathbf{z})$.

**Practical approach and algorithm details.** For warmstarting the abstraction generator, we utilize abstractions generated by o4-mini. We then use a weaker model (GPT-4.1-mini) to evaluate each abstraction by comparing solution success with and without it, retaining only those that improve performance to form our seed set. Then, we run SFT of Qwen3-1.7B for 5 epochs on this seed dataset to obtain our initial abstraction generator (Qwen Team, 2025). For the solution generator, we use the same Qwen3-1.7B model, ensuring that both components use models of identical capacity.

After SFT, we employ `RLAD` to fine-tune the abstraction generator and abstraction-conditioned solution generator via RL. For the abstraction generator, we opt to use "batched" offline RL via

RFT (Yuan et al., 2023) and RPO (Pang et al., 2024), since reward computation by rolling out the solution generator the on the fly and running online RL was infeasible using our RL infrastructure and within compute we had access to. To train the solution generator, we utilize the DAPO approach (Yu et al., 2025), and include token-level policy loss normalization and asymmetric clipping, and prompt difficulty/length curriculum. Building upon implementation of concurrent work (Setlur et al., 2025), we employ a two stage curriculum where we partition the DeepScaleR (Luo et al., 2025) mixture by success rate of the base model into three sets: (1) easy, (2) medium, and (3) hard, where we fine-tune first on easy problems with an 8K token budget and then on medium problems. We utilize the hard split as a held out, evaluation subset, which we denote as "**DeepScaleR [Hard]**". We outline hyperparameters and details in Appendix A.1 and provide a pseudocode in Algorithm 1.

---

**Summary: RLAD method design**

RLAD jointly optimizes the abstraction generator $\pi_\theta^{\mathrm{abs}}$ and solution generator $\pi_\theta^{\mathrm{sol}}$ with RL, using reward functions in Equation 3. These reward functions incentivize $\pi_\theta^{\mathrm{sol}}$ to utilize abstractions and incentivize $\pi_\theta^{\mathrm{abs}}$ to propose useful abstractions per problem.

---

## 6 EXPERIMENTAL EVALUATION

The goal of our experiments is to evaluate the efficacy of RLAD in improving the reasoning capabilities of LLMs through abstraction-guided solution generation. Specifically, we aim to answer the following research questions: **(1)** Does RLAD improve pass@1 accuracy across several reasoning benchmarks compared to direct solution generation?, **(2)** How does RLAD scale as more abstractions and solutions are generated?, and **(3)** What makes the generated abstractions useful, how faithfully are they followed, and how do they guide and improve solution generation? We compare RLAD with strong models on three math reasoning benchmarks: AMC 2023, AIME 2025, and DeepScaleR Hard (Luo et al., 2025), which itself is a subset of hard problems from the OmniMATH mixture on which DeepSeek-R1 distilled Qwen-32B model attains an accuracy of $\leq 10\%$. We also fine-tune an abstraction generator for the ARC-AGI program synthesis tasks, and conduct a similar comparison on 90 ARC puzzles evenly derived from the test sets of ARC-AGI 1, ARC-AGI 2, and BARC (Li et al., 2024). We also perform several ablations to better understand abstractions produced by RLAD.

### 6.1 MAIN PERFORMANCE RESULTS ON MATH REASONING BENCHMARKS

We evaluate RLAD under three settings: (1) **w/o abs**, without abstractions; (2) **w/ abs (avg)**, average performance over generations conditioned on 4 proposed abstractions per problem; and (3) **w/ abs (best)**: using the best-performing abstraction (in a set of 4 proposed abstractions per problem). We observe that RLAD outperforms the base model and variant fine-tuned with RL on the same prompts via DAPO (Yu et al., 2025) without any abstractions, across all settings and benchmarks (Table 2). This highlights that RLAD can propose and leverage abstractions to improve reasoning performance. Interestingly, we also note that these performance gains are not limited to abstraction-conditioned inference: even in the **w/o abs** setting, where no abstraction is provided during inference, RLAD improves over the prior methods, when trained with abstractions via RLAD. This suggests that exposure to diverse abstractions during training enhances the model's general reasoning ability. We observe similar trends on additional benchmarks, including AIME 2024 and HMMT 2025 (see Appendix B.1), where RLAD improves in the w/o abs setting. Finally, in Appendix D, we also measure the performance of RLAD when different token budgets are allowed for reasoning – while Table 2 measures performance at a budget of 32K tokens, we also measure performance at 8K and 16K budgets and find RLAD to be more effective than other approaches.

| Approach | AIME 2025 | | | DeepScaleR [Hard] | | | AMC 2023 | | |
|---|---|---|---|---|---|---|---|---|---|
| | w/o abs (avg) | w/ abs (avg) | w/ abs (best) | w/o abs (avg) | w/ abs (avg) | w/ abs (best) | w/o abs (avg) | w/ abs (avg) | w/ abs (best) |
| Qwen-3-1.7B | 33.75 | 36.25 | 40.00 | 20.21 | 22.14 | 32.50 | 86.41 | 78.01 | 84.53 |
| + DAPO | 37.92 | 34.90 | 39.79 | 21.67 | 21.88 | 33.54 | 86.41 | 81.99 | 88.44 |
| + **RLAD (Ours)** | 38.04 | **42.45** | **48.33** | 23.54 | **24.84** | **35.54** | 87.25 | **88.35** | **91.72** |

Table 2: **Accuracy on math reasoning benchmarks.** RLAD achieves consistent gains in abstraction-conditioned and w/o abstractions. Here, we measure performance on 3 domains (AIME 2025, DeepScaleR Hard, and AMC 2023) with the base Qwen 3-1.7B model, DAPO, and RLAD. We measure performance without abstractions, with abstractions (pass@1 with 16 samples) and the best abstraction (pass@16), for each method type.

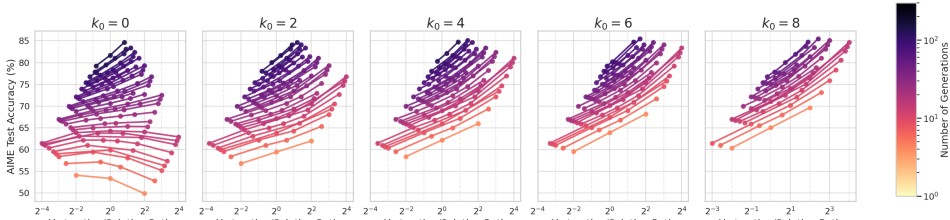

Figure 5: *Tradeoff of abstraction and solution generation on AIME 2025.* As the total inference compute budget increases (color scheme on the right), we find better performance efficiency when allocating our budget to abstraction generation rather than solution generation, for all values of normalization offset $k_0$ given to us.

## 6.2 UNDERSTANDING PROPERTIES OF RLAD

In this section, we conduct a number of additional experiments to understand the behavior of RLAD in regards to the usefulness of the proposed abstractions and algorithm performance when provided with a large budget on the total inference compute.

**1) "Weak-to-strong" generalization of the abstraction generator.** We next evaluate the weak-to-strong generalization of our method by pairing our trained abstraction generator with o4-mini as the solution generator. We use a fixed 24k generation token budget and 4 samples per question. Without abstractions, o4-mini achieves 80.38% pass@1, 82.26% pass@2, and 84.77% pass@4. Conditioning on both the problem and the proposed abstractions improves performance to 85.83% pass@1, 88.33% pass@2, and 90.00% pass@4 accuracy. Thus, conditioning on abstractions consistently yields higher pass@k accuracy compared to question-only conditioning, even for this strong reasoning model. These gains demonstrate that abstractions, though produced by a comparatively weaker model, can transfer effectively to a stronger solution generator, providing evidence that abstractions generalize in a weak-to-strong setting and enables downstream improvements without additional supervision or modifications to the strong model.

**2) Compute tradeoffs b/w abstraction and solution generation.** Next, we study how to allocate compute between generating more abstractions and sampling solutions to attain maximal performance within a given inference budget.

|  | 1 | 4 | 16 | 64 | 256 |
|---|---|---|---|---|---|
| DAPO, w/o abs (pass@$n^2$) | 0.37 | 0.51 | 0.65 | 0.77 | 0.82 |
| RLAD, w/ abs (pass@$n \times n$) | **0.41** | **0.59** | **0.71** | **0.80** | **0.87** |

Table 3: **Pass@k comparison under equal compute.** For each $n$, we compare $n^2$ solution samples (no abstractions) with $n$ abstractions $\times$ $n$ solutions per abstraction. Abstraction conditioning yields consistent improvements.

This corresponds to a "compute-optimal strategy" (Snell et al., 2024) for partitioning compute between abstraction and solution generation. If the model typically fails by making small local errors in its computation, then additional concise abstractions may not help it as much as simply trying again to generate a locally similar solution (optimizing "depth"). In contrast, if the model tends to pursue a seemingly plausible but incorrect approach and is unable to easily recover or switch approaches, then conditioning on diverse abstractions can help by offering alternative high-level approaches toward the correct answer. In other words, when the model has a tendency to explore "depth" over "breadth" of solution strategies, abstractions can help improve performance. With this intuition, we hypothesize that when the compute budget permits only a limited number of samples, allocating more compute to sampling multiple solutions but retaining only a few abstractions will improve success more. However, once pass@k for a single abstraction begins to saturate, performance gains are more likely to come from scaling the diversity of abstractions, which enables the model to explore qualitatively different regions of the solution space.

To validate this hypothesis, we plot *iso-compute* scaling curves under a fixed compute budget $\mathcal{C}$, we distribute between abstractions and solution generation. Specifically, we denote the number of abstractions as $m$ and the number of solutions sampled per abstraction as $k$, such that $m \times k = \mathcal{C}$. To better isolate the utility of abstractions, we make several more projections of the iso-compute curve, each with a different "normalization offset" $k_0$. This normalization offset $k_0$ *discounts* performance gains that stem from trying the problem again and making local modifications (e.g., small edits that do not require new strategy changes), but measures the performance gains that stem from major changes in content of a response. Concretely, for a non-zero $k_0$, we plot *iso-compute* frontiers when $m \times (k - k_0) = \mathcal{C}$. This formulation captures the amount of compute that is spent on "meaningful"

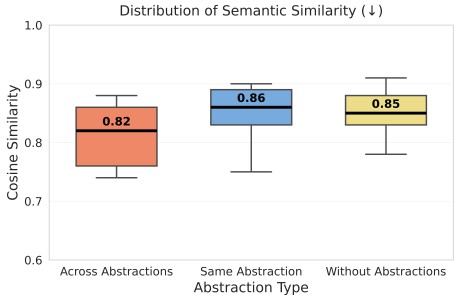 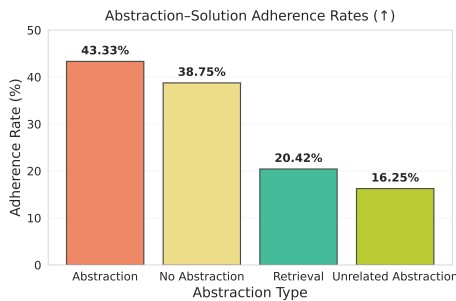

Figure 6: **Abstraction-conditioned solution generation analysis.** RLAD produces solutions with **(left)** greater semantic diversity across different abstractions and **(right)** higher abstraction adherence than baselines.

samples that go beyond the model's local neighborhood. Figure 5 shows these *iso-compute* frontiers for different values of the compute budget $\mathcal{C}$. The x-axis plots the ratio between abstractions and adjusted solutions, $m/(k - k_0)$. Each curve corresponds to a fixed total compute budget.

In Figure 5, across $k_0 \in \{0, 2, 4, 6, 8\}$, shifting compute toward abstractions consistently yields greater improvements than allocating the same additional compute to solution refinements, especially as the total compute budget increases. ***This supports the conclusion that*** *once local errors in the chain-of-thought have been addressed, it is more effective to increase the diversity of strategies used via abstraction conditioning rather than to continue to scale up long CoT sampling alone.*

To obtain additional evidence to support this result, we also evaluate pass@k of DAPO and RLAD for a fixed compute budget for both abstraction and non-abstraction generation on AIME 2025. Here, for a given value $n$, we compare two approaches: **(1)**sampling $n^2$ solutions without abstractions, and **(2)** sampling $n$ abstractions and then $n$ solutions per abstraction. In Table 3, abstraction conditioning consistently outperforms pure solution sampling. For example, at $n = 16$, abstraction conditioning achieves a pass@k of $0.71$ versus $0.65$ without abstractions, and at $n = 256$ the gap widens to $0.87$ versus $0.82$. This corroborates the efficacy of abstractions, even under matched compute budgets.

*3) Understanding the behavior of the abstraction-conditioned solution generator.* A desirable property of the solution generator is the ability to follow the proposed abstractions. To study this, we prompt (o4-mini) to classify whether a particular solution trace produced by a trained solution generator closely adheres to a given abstraction. We ask for a binary decision on each pair of abstraction and solution, and measure the adherence rate across 200 such pairs. In Figure 6 (right), we report the adherence rates under four conditions: **(1)** Abstraction, where we measure adherence rates between an abstraction and a solution generated by conditioning on this abstraction itself; **(2)** No abstraction, where we measure the adherence rates between an abstraction and a solution generated by conditioning on the problem without any abstraction; **(3)** Retrieval, where we measure adherence rates between an abstraction and a semantically similar prior solution to the problem; and **(4)** Unrelated abstraction, where we measure the adherence rates between an abstraction and a solution generated via a different abstraction. We find that the Abstraction condition achieves the highest adherence rate. Intuitively, this means that the trained solution generator is detected to be more likely to follow the strategy or guidance of a given abstraction. Additionally, we measure the semantic similarity of solutions generated without abstraction conditioning, conditioned on the same abstraction, and across abstractions and make a similar conclusion.

> **Takeaways: Experimental Results**
>
> RLAD outperforms RL fine-tuning approaches that do not propose or leverage abstractions on math reasoning. Jointly scaling the number of abstractions and solution samples enables continued performance gains even when scaling solutions alone begins to saturate.

## 7 DISCUSSION AND PERSPECTIVES ON FUTURE WORK

We introduce reasoning abstractions: concise natural language representations of procedural and factual knowledge, as a way to expand LLM reasoning strategies. Our method, RLAD, jointly trains an abstraction generator and an abstraction-conditioned solution generator, achieving consistent gains on mathematical reasoning benchmarks. We show that allocating compute to diverse abstractions yields larger improvements than simply increasing solution sampling, offering a complementary axis for scaling test-time compute. While our study focuses on math tasks, extending abstractions to broader reasoning and unifying abstraction and solution generation remain open directions.

## 8 ETHICS STATEMENT

Our empirical evaluation focuses primarily on mathematical reasoning (e.g., AIME, AMC, Deep-ScaleR) and abstract pattern recognition (ARC-AGI). These domains are generally considered low-risk, as they utilize publicly available benchmarks that do not contain private or sensitive personal data. The risk of generating toxic or harmful content in these structured reasoning contexts is minimal compared to open-ended generation tasks. However, we acknowledge the broader ethical considerations associated with advancing LLM capabilities. While enhanced reasoning abilities offer significant potential for scientific and educational benefits, they also carry dual-use risks if applied to malicious activities, such as generating sophisticated disinformation. We encourage the continued development of robust safety and alignment protocols alongside improvements in reasoning capabilities, which are jointly correlated as evidenced in prior work (Kim et al., 2025). Finally, we utilized LLMs such as GPT5/Gemini for minor rewritings throughout the paper for better readability.

## 9 REPRODUCIBILITY STATEMENT

We are committed to ensuring the transparency and reproducibility of our research. To facilitate the replication of our results, we have provided comprehensive details regarding our methodology and experimental setup in the Appendix and main paper. In particular, the RLAD framework, including the two-player RL training paradigm and the specific reward design (Equations 1, 2 and 3), is detailed in Section 5. The procedure for warmstarting the models via Supervised Fine-Tuning (SFT), including the generation of seed abstractions using stronger models (o4-mini and GPT 4.1-mini) and the filtering process, is described in Section 5. The implementation details of the curriculum learning strategy are also provided in Section 5 and analyzed in Appendix B. Additionally, detailed implementation information is provided in the Appendix. Here, the pseudocode for the joint RL training process is available in Appendix A.1 (Algorithm 1). All key training hyperparameters used for the RL methods (DAPO, RFT, RPO) are listed in Appendix A.1 (Table 4). Our experiments utilize publicly available base models from the Qwen3 series (Qwen Team, 2025). The evaluations are conducted on standard reasoning benchmarks: AIME 2025 (Mathematical Association of America, 2025), DeepScaleR (Luo et al., 2025), AMC 2023 (Zeng et al., 2025), and ARC-AGI (Li et al., 2024), as described in Section 6, with additional results in Appendix B. Qualitative examples and the prompt used for abstraction classification are included in Appendix D.

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

# Appendices

## A  Experimental Details

### A.1  Pseudocode for RLAD

---
**Algorithm 1** Joint RL Training of $\pi_\theta^{\mathrm{abs}}$ and $\pi_\theta^{\mathrm{sol}}$

---
**Require:** Policies $\pi_\theta^{\mathrm{abs}}(\mathbf{z} \mid \mathbf{x})$, $\pi_\theta^{\mathrm{sol}}(\tilde{\mathbf{y}} \mid \mathbf{x}, \mathbf{z})$ Datasets $\mathcal{D}_{\pi_\theta^{\mathrm{abs}}}$, $\mathcal{D}_{\pi_\theta^{\mathrm{sol}}}$; rates $\alpha_{\pi_\theta^{\mathrm{abs}}}, \alpha_{\pi_\theta^{\mathrm{sol}}}$; batch sizes $N, M$; epochs $E$

1:  Initialize $\pi_\theta^{\mathrm{abs}}, \pi_\theta^{\mathrm{sol}}$
2:  **for** $e = 1$ to $E$ **do**                                          ▷ Update abstraction policy
3:      **for** $\{\mathbf{x}_i\}_{i=1}^{N} \sim \mathcal{D}_{\pi_\theta^{\mathrm{abs}}}$ **do**
4:          $\mathbf{z}_i \sim \pi_\theta^{\mathrm{abs}}(\cdot|\mathbf{x}_i)$
5:          $r_i \leftarrow r_{\pi_\theta^{\mathrm{sol}}}(\mathbf{x}_i, \mathbf{z}_i)$
6:          $\pi_\theta^{\mathrm{abs}} \leftarrow \pi_\theta^{\mathrm{abs}} - \alpha_{\pi_\theta^{\mathrm{abs}}} \nabla_{\pi_\theta^{\mathrm{abs}}} \mathcal{L}_{\mathrm{STAR/RPO}}(\pi_\theta^{\mathrm{abs}}; \mathbf{x}_i, \mathbf{z}_i, r_i)$
7:      **end for**                                                       ▷ Update solution policy
8:      **for** $\{\mathbf{x}_j\}_{j=1}^{M} \sim \mathcal{D}_{\pi_\theta^{\mathrm{sol}}}$ **do**
9:          $\mathbf{z}_j \sim \pi_\theta^{\mathrm{abs}}(\cdot|\mathbf{x}_j), \quad \tilde{\mathbf{y}}_j \sim \pi_\theta^{\mathrm{sol}}(\cdot|\mathbf{x}_j, \mathbf{z}_j)$
10:         $r_j \leftarrow r(\mathbf{x}_j, \mathbf{z}_j, \tilde{\mathbf{y}}_j)$
11:         $\pi_\theta^{\mathrm{sol}} \leftarrow \pi_\theta^{\mathrm{sol}} - \alpha_{\pi_\theta^{\mathrm{sol}}} \nabla_{\pi_\theta^{\mathrm{sol}}} \mathcal{L}_{\mathrm{GRPO}}(\pi_\theta^{\mathrm{sol}}; \mathbf{x}_j, \mathbf{z}_j, \tilde{\mathbf{y}}_j, r_j)$
12:     **end for**
13: **end for**

---

### A.2  Hyperparameters

| Hyperparameter | Value |
|---|---|
| algorithm | DaPO (Yu et al., 2025) |
| training steps | 100 |
| epochs | 10 |
| train batch size | 128 |
| max prompt length | 3072 |
| max response length | 16384 |
| max extrapolation length | 32768 |
| learning rate | 1e-6 |
| clip ratio (low / high) | 0.2 / 0.5 |
| entropy coefficient | 0.001 |
| KL loss coefficient | 0.001 |
| KL loss type | low_var_kl |
| sampling temperature (train / val) | 0.6 / 0.6 |
| samples per prompt (train / val) | 16 / 8 |
| max batched tokens | 32768 |

Table 4: Key training hyperparameters used in RLAD.

## B  Additional Experimental Results

### B.1  RLAD's w/ abs performance on AIME 2024 and HMMT 2025

In this section, we evaluate the performance of the base model (Qwen-3-1.7B), GRPO-enhanced model, and our proposed method RLAD on two math reasoning benchmarks: AIME 2024 and HMMT 2025. As shown in Table 5, our method achieves the best performance across both datasets.

It is important to note that RLAD is trained using access to abstractions, yet it also generalizes better even when evaluated without abstraction. This suggests that RLAD does not merely overfit to the

abstraction format but instead learns to effectively leverage high-level procedural guidance, leading to better generalization on challenging reasoning benchmarks.

| Approach | AIME 2024 | HMMT 2025 |
|---|---|---|
| Qwen-3-1.7B | 48.54 | 22.50 |
| + DaPO | 44.17 | 23.13 |
| + `RLAD` | **51.46** | **23.75** |

Table 5: **`RLAD`'s w/ abs performance on AIME 2024 and HMMT 2025.** `RLAD` outperforms both DaPO and the base Qwen model on the AIME 2024 and HMMT 2025 benchmarks.

## B.2 DESIGN CHOICE ABLATIONS

In this section, we conduct ablation studies to isolate the contributions of key components in `RLAD`. We investigate three primary design choices, with results summarized in Table 6: **(a)** utilizing a curriculum training strategy, **(b)** including prompts not annotated with an abstraction, and **(c)** applying reward masking to those non-annotated prompts.

**Curriculum training** refers to a staged process where the model first learns from simpler problems and gradually transitions to harder ones. We use the protocol from Setlur et al. (2025) as inspiration, who demonstrated its effectiveness for direct math problem-solving. In our setting, which incorporates abstractions, curriculum training also proves beneficial, improving both average and best-case performance from 0.38 and 0.43 to 0.41 and 0.48, respectively, compared to non-curriculum training.

Next, we analyze the practice of **including prompts without abstractions** and applying **reward masking**. Including a small fraction of these "no-abstraction" prompts is intended to better condition the solution-generator on the abstractions when they are present. However, this risks the model learning a shortcut by simply ignoring the abstractions. To mitigate this, we apply reward masking: for completions on no-abstraction prompts, we nullify the policy reward by zeroing out the advantage, while retaining the KL penalty for regularization. This prevents the model from over-optimizing on examples that lack abstractions, a behavior that would otherwise hinder generalization.

Our findings confirm the efficacy of this combined approach. As shown in Table 6, including no-abstraction prompts with reward masking is critical for performance. Ultimately, the combination of all three design choices—curriculum training, the inclusion of no-abstraction prompts, and reward masking—significantly outperforms alternative configurations.

| Approach | Design Choice | | | AIME 2025 | |
|---|---|---|---|---|---|
| | curriculum training | including no-abstraction prompt | reward masking | w/ abs (avg) | w/ abs (best) |
| variant 1 | ✗ | ✓ | ✗ | 36.51 | 42.29 |
| variant 2 | ✗ | ✗ | - | 37.08 | 42.50 |
| variant 3 | ✗ | ✓ | ✓ | 37.50 | 43.33 |
| `RLAD` | ✓ | ✓ | ✓ | **42.45** | **48.33** |

Table 6: **Design Choices in `RLAD`.** We isolate the effects of curriculum training, no-abstraction inclusion, and reward masking. The full method achieves the strongest performance under abstraction-conditioned evaluation.

## B.3 FULL RESULTS FOR ABSTRACTIONS IN NON-MATH DOMAINS

As seen in Table 7, conditioning on abstractions helps for 37 domains, where the average and best abstractions outperform standard prompting by 18.0% and 30.0% on average, respectively.

For ARC-AGI, we warmstart the abstraction generator model with synthetically augmented human annotations from the BARC dataset (Li et al., 2024). In Table 1, we report results using an abstraction generator and a solution generator instantiated as Qwen3-4B.

## C ABSTRACTIONS IN OTHER DOMAINS

## D QUALITATIVE EXAMPLES OF MATH REASONING ABSTRACTIONS

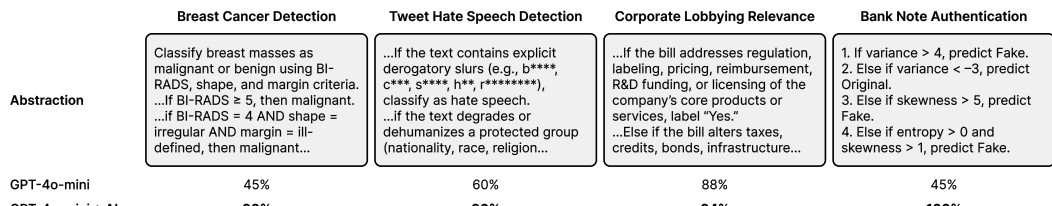

| | Breast Cancer Detection | Tweet Hate Speech Detection | Corporate Lobbying Relevance | Bank Note Authentication |
|---|---|---|---|---|
| Abstraction | Classify breast masses as malignant or benign using BI-RADS, shape, and margin criteria. ...If BI-RADS ≥ 5, then malignant. ...If BI-RADS = 4 AND shape = irregular AND margin = ill-defined, then malignant... | ...If the text contains explicit derogatory slurs (e.g., b****, c***, s****, h**, r********), classify as hate speech. ...if the text degrades or dehumanizes a protected group (nationality, race, religion... | ...If the bill addresses regulation, labeling, pricing, reimbursement, R&D funding, or licensing of the company's core products or services, label "Yes." ...Else if the bill alters taxes, credits, bonds, infrastructure... | 1. If variance > 4, predict Fake. 2. Else if variance < −3, predict Original. 3. Else if skewness > 5, predict Fake. 4. Else if entropy > 0 and skewness > 1, predict Fake. |
| GPT-4o-mini | 45% | 60% | 88% | 45% |
| GPT-4o-mini + Abs | **90%** | **90%** | **94%** | **100%** |

Figure 7: *Examples of good reasoning abstractions in non-math domains*. Adding the abstraction to the prompt of GPT-4o-mini consistently improves performance on unseen instances.

*Interpreting discovered abstractions.* As discussed in Appendix D, we classify each model-generated abstraction into four categories for ease of interpretability: (1) **caution alert** that warns the solution generator to avoid a specific approach; (2) **productive launchpoint** that suggests strategic framings or problem reformulations that open high-potential solution paths; (3) **blind-follow trajectory** that prescribes repeatable, step-by-step procedures executable without furtheАr insight; and (4) **structural shortcut** that leverages abstract insights or invariants to collapse multiple reasoning steps into a single leap. In Figure 8, we show that after training via RLAD, the distribution over these categories

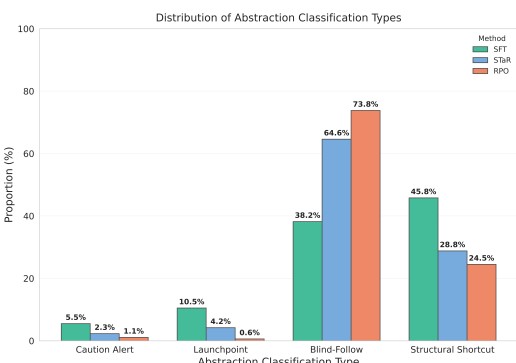

Figure 8: **Abstraction Categorization** RLAD produces a diverse characterization of abstractions, which we characterize by prompting o4-mini.

shifts, with a notable increase in blind-follow abstractions, which a stronger reasoning model classifies as an effective reasoning path to a successful solution as seen in Appendix D.

## D.1 PROMPT FOR ABSTRACTION CLASSIFICATION

We prompt GPT-4o-mini with the following prompt template to classify each abstraction into one of four categories.

```
Post-hoc abstraction classifier prompt

You are a abstraction classifier.  You will be given a
problem-solving heuristic or abstraction used for mathematical
reasoning.  Your task is to classify it into exactly one of the
following mutually exclusive categories, based on the primary
cognitive function the heuristic serves.

(A) Caution alert:  any abstraction that warns the reader to
double-check a specific aspect of their solution or to not take a
specific approach to the problem.
(B) Productive launchpoint:  an early move or framing that
opens up high-potential trajectories.  Examples include clever
reformulations or symmetries.
(C) Blind-follow trajectory:  a description of a repeatable,
sequential path that can be reliably followed to solve the problem.
Examples include plug-and-play formulas that can be followed
blindly, without insight.  Do not choose this is further reasoning
is required to solve the problem.
(D) Structural shortcut:  a conceptual move that collapses multiple
graph paths into a single jump via insight or abstraction.  This
can include introducing invariants.
(E) Other:  a abstraction that does not fit into the above
categories.
Give a 1-2 sentence explanation for your classification, and end
your answer with exactly one of:  (A), (B), (C), (D), or (E).
```

```
--
abstraction:
{abstraction}
```

## D.2 EXAMPLE FOR SHORT ABSTRACTION

> **Examples of Short Abstractions**
>
> **Question:** The banker's gain of a certain sum due 3 years
> hence at 10% per annum is Rs. 36. What is the present worth?
> **Hint:**
> – Consider the proportional relationship between the true
> discount and the present worth, and how the banker's gain
> relates to these terms.
> – Think of the banker's gain as a component of the true
> discount and how it relates to the present worth through the
> interest rate and time period.
>
> **Question:** A train 125 m long passes a man running at 15
> km/hr in the same direction in which the train is going, in
> 10 seconds. What is the speed of the train?
> **Hint:**
> – Think about the speed difference between the train and the
> man, and how that difference relates to the time it takes for
> the entire train to pass by.
> – Consider the difference in speeds as the key factor in
> determining the train's speed relative to the man, and use
> the time taken to pass the man to find this difference.

## D.3 EXAMPLE FOR EACH ABSTRACTION CATEGORY

Below, we show examples of abstractions classified into the four categories above.

> **Examples of (A) Caution alert**
>
> <description>Always record forbidden values from denominators before
> and after manipulation. After solving the polynomial, discard
> any roots that make a denominator zero or that do not satisfy the
> original equation, to avoid extraneous solutions.</description>
> <example>In the equation (x+2)/(2x−1) = x−3, 2x−1 cannot be zero
> (so x is not ½). If solving yields x=½ or any root that makes any
> denominator zero, reject it. Then verify the accepted roots in the
> original equation.</example>
>
> <description>Keep units consistent when moving between area and
> length or summing lengths. After extracting a length from an area
> (via square root), ensure subsequent arithmetic stays in the same
> unit to avoid scaling errors. </description>
> <example>If a square's area is 10000 cm², its side is sqrt(10000) =
> 100 cm. To express in meters, convert 100 cm to 1 m. All later
> distances computed with that side length must be in meters to
> remain consistent.</example>

> **Examples of (B) Productive launchpoint**
>
> <description>Translate comparative statements into algebraic
> equations using the chosen variables. Phrases like "twice
> as many" or "one less than" correspond to multiplication or

```
addition/subtraction expressions.  This step captures the core
relationship in a solvable form.</description>
<example>If the problem states "Group A has twice as many as Group
B," write the equation x = 2y.  For "Group B has three fewer than
Group C," you would write y = z - 3.</example>

<description>Select one variable as a parameter (often setting it
to 1 or keeping it symbolic) to express all other variables in
terms of it.  This reduces the number of independent symbols and
streamlines substitutions.</description>
<example>Given p/q = 3 and r/q = 2, choose q as the base variable.
Write p = 3q and r = 2q, so all expressions involving p and r can
be handled through q alone.</example>
```

## Examples of (C) Blind-follow trajectory

```
<description>Logarithms offer a streamlined way to compute
floor-based digit counts:  for y>0, the number of integer digits is
floor(log10 y) + 1.  Use this to handle arbitrary exponents without
juggling large powers explicitly.</description>
<example>To count digits of y = $x^7$, compute d = floor(7 * log10
x) + 1.  If x=2.5, then d = floor(7 * log10(2.5))+1 = 2+1 = 3
digits.</example>

<description>The mean of a set equals its total sum divided by its
number of elements.  Use this to move between sums and averages
when counts or totals are known.  It works because "average" is
defined as that ratio.</description>
<example>Suppose a subset has k items with mean m.  Then its total
sum is S = k·m.  Conversely, if you know the sum S and the count k,
the mean is m = S/k.  For instance, if 5 items average to 10, their
total is 5×10 = 50, and if you later learn the total is 60 for 6
items, the new mean becomes 60/6 = 10.</example>
```

## Examples of (D) Structural shortcut

```
<description>When the same distance appears in multiple geometric
roles (e.g., as radius to a vertex and to a tangen t point),
express it in different algebraic forms and equate them.  Solving
the resulting equation produces the unknown variable, which then
gives the desired length.</description>
<example>If r is both the distance from O to a vertex (r = sqrt[x² +
(L/2)²]) and the distance from O to the tangent point (r = f(x)),
set sqrt[x² + (L/2)²] = f(x).  Solving this equation for x and
back-substituting determines r explicitly, closing the geometric
problem with an algebraic solution.</example>

<description>Use the perimeter constraint a+b+c=P to eliminate one
variable, e.g.  set c=P-a-b, reducing the problem to two degrees
of freedom.  This simplification turns the three-variable Heron
expression into a function of a and b alone, facilitating analysis
or enumeration.</description>
<example>For a target perimeter P=10, one writes c=10-a-b.
Substituting into Heron's formula yields A(a,b)=sqrt[5 * (5-a) *
(5-b) * (a+b-5)], which is now a two-variable function to study
instead of three.</example>
```

| Dataset | Zero-shot | Best Abstraction | Average Abstraction |
|---|---|---|---|
| UCI Dry Bean | 0.00 | 0.65 | 0.51 |
| Wikipedia Proteinogenic Acid | 0.22 | 0.78 | 0.58 |
| UCI Student Performance | 0.25 | 0.45 | 0.28 |
| UCI Website Phishing | 0.25 | 0.25 | 0.22 |
| UCI Teaching Assistant Evaluation | 0.25 | 0.45 | 0.33 |
| UCI Contraceptive Method Choice | 0.30 | 0.60 | 0.43 |
| UCI Vertebral Column | 0.30 | 0.75 | 0.64 |
| UCI Shill Bidding | 0.30 | 1.00 | 0.95 |
| Kaggle Job Change | 0.30 | 0.85 | 0.83 |
| UCI Caesarian Section | 0.38 | 0.75 | 0.64 |
| Wikipedia Coin Face Value | 0.40 | 1.00 | 0.88 |
| UCI Wine | 0.40 | 0.95 | 0.85 |
| UCI Tic-Tac-Toe Endgame | 0.40 | 0.80 | 0.42 |
| Kaggle Campus Placement | 0.40 | 0.85 | 0.72 |
| Wikipedia Driving Championship Points | 0.40 | 1.00 | 0.74 |
| UCI Mammographic Mass | 0.45 | 0.90 | 0.82 |
| UCI Banknote Authentication | 0.45 | 1.00 | 0.78 |
| Kaggle Engineering Placement | 0.50 | 0.85 | 0.79 |
| RAFT One Stop English | 0.50 | 0.40 | 0.36 |
| LegalBench Function of Decision Section | 0.54 | 0.72 | 0.61 |
| Kaggle Entrepreneur Competency | 0.55 | 0.65 | 0.58 |
| UCI Indian Liver Patient | 0.55 | 0.80 | 0.68 |
| LegalBench International Citizenship Questions | 0.56 | 0.74 | 0.63 |
| LegalBench Abercrombie | 0.56 | 0.80 | 0.67 |
| Wikipedia Color Luminance | 0.60 | 1.00 | 1.00 |
| RAFT Twitter Hate Speech | 0.60 | 0.90 | 0.76 |
| Wikipedia Award Nomination Result | 0.64 | 1.00 | 0.76 |
| UCI Car Evaluation | 0.65 | 0.75 | 0.64 |
| Kaggle Water Potability | 0.65 | 0.50 | 0.38 |
| Kaggle Travel Insurance | 0.65 | 0.70 | 0.59 |
| UCI Internet Firewall | 0.70 | 1.00 | 0.97 |
| RAFT ADE Corpus | 0.70 | 1.00 | 0.89 |
| UCI Somerville Happiness Survey | 0.70 | 0.80 | 0.68 |
| UCI Mushroom | 0.75 | 1.00 | 0.95 |
| UCI Occupancy Detection | 0.80 | 1.00 | 0.92 |
| Kaggle Stroke Prediction | 0.85 | 0.90 | 0.90 |
| LegalBench Corporate Lobbying | 0.88 | 0.94 | 0.88 |
| Average | 0.50 | 0.80 | 0.68 |

Table 7: Evaluation of abstractions on diverse collection of 37 domains. We sampled 10 abstractions by prompting `o4-mini`, and measure test set accuracy while prompting `GPT-4o-mini` with each abstraction. We report both the average performance of the 10 abstractions and the best abstraction. **We find that the average and best abstractions outperform standard prompting by 18.0% and 30.0% on average, respectively.**

### D.4 THE USE OF LARGE LANGUAGE MODELS (LLMs)

Large Language Models are used to assist with proofreading and minor wording improvements. All research ideas, experiments, and conclusions were conceived and validated by the authors. Additionally, tools such as Cursor were utilized as coding assistants during the development of the coding infrastructure for the project.

