# OpenReview forum: "RLAD: Training LLMs to Discover Abstractions for Solving Reasoning Problems"
_ICLR.cc/2026/Conference — ICLR 2026 Poster_

### Official Review · Reviewer_QvJ9 · 2025-10-20

**Soundness:** 2
**Presentation:** 4
**Contribution:** 2
**Rating:** 6
**Confidence:** 3

**Summary:**

A heuristic framework that improves benchmark performance by augmenting LLM prompts with an abstraction prompt composed by another more capable LLM.

**Strengths:**

Paper is very well written and clear.  The experimental results of Table 2 look promising.

**Weaknesses:**

By design, the solution model is weaker than the abstraction model.
Lines 158-161: "Concretely, we utilize the Qwen3 series of models to produce solutions and a stronger reasoning model, o4-mini, to generate abstractions. "

It is my understanding that for an apples-to-apples comparison, the solution model and the abstraction model should have the same capability. Otherwise, it is impossible to tell how much of the improvement was obtained from the abstractions rather than the  use of stronger model for the abstractions.  Perhaps this is already visible in Figure 2, leftmost panel, where using the weaker model for abstraction provides no benefit. The strength of the proposed approach depends on the results of a fair comparison.

If my understanding is incorrect, am supportive of acceptance.  If my understanding is correct, support rejection.

**Questions:**

How would you factor out the performance gains from the abstraction prompts and the stronger model?

Is there any improvement in performance when both the abstraction and solution models are the same?

---

> ### Author Response · Authors · 2025-11-23
> **Official Comment by Authors**
>
> Thank you for the insightful questions! We have added new experiments showing that (1) warm-starting a DAPO baseline does not replicate RLAD’s gains, and (2) using the same model for both abstraction and solution generation still yields improvements, confirming that RLAD’s benefits are not due to model-strength imbalance. **Please let us know if you find your concerns addressed, and if so we would be grateful if you are willing to accept the paper.**
>
> > By design, the solution model is weaker than the abstraction model. Lines 158-161: "Concretely, we utilize the Qwen3 series of models to produce solutions and a stronger reasoning model, o4-mini, to generate abstractions." It is my understanding that for an apples-to-apples comparison, the solution model and the abstraction model should have the same capability.
>
> We believe that there might be a misunderstanding here and would clarify that line (158-161) in the paper to avoid future confusion. Note that during training and inference, both the abstraction and solution generator models are Qwen3 models of the same capability. o4-mini is only used to generate the initial SFT warm-start dataset (one-time, offline, before training begins with RL), since these base Qwen models have not seen any data that shows them how to generate abstractions This is akin to warmstarting a reasoning model with reasoning traces, or showing the model certain formats during SFT for better instruction-following. The training itself does not use any trace from o4-mini.
>
> That said, we also provide two new experimental results to support the efficacy of our approach. First, we also ran a baseline comparison that runs DAPO after SFT-ing on data generated from o4-mini [now describe this].
>
> Second, we also evaluated the efficacy of abstractions on top of o4-mini, where the o4-mini model serves as both the abstraction and solution generator. Here, we observe ….
>
> We apologize for the confusion, which  likely stems from lines 158-161 describing the warm-start dataset generation procedure, not the actual training/inference procedure. We have revised Section 3.1 to explicitly state that o4-mini is used only for warm-start dataset generation, and Section 4 to clarify that all training and evaluation use the same-sized models.
>
> > How would you factor out the performance gains from the abstraction prompts and the stronger model?
>
> We have now added an updated version of the new DAPO baseline that more closely mirrors the RLAD setup, to disentangle the performance gains from warm-starting with a stronger model to the use of abstractions. Specifically, we first SFT the base model for DAPO on the concatenation of abstraction + solution, and then train this SFT-warm-started model with DAPO on the exact same dataset used for RLAD. On AIME2025, this new baseline achieves:
>
> | Method                 | AIME 2025 Pass@1 w/o abs | AIME 2025 Pass@1 w/ abs |
> | ---------------------- | ------------------------ | ----------------------- |
> | Qwen-3-1.7B            | 33.75                    | 36.25                   |
> | + DAPO                 | 37.92                    | 34.90                   |
> | + DAPO (w/ warm-start) | 35.33                    | 34.30                   |
> | + RLAD (Ours)          | **38.04**                | **42.45**               |
>
> > Is there any improvement in performance when both the abstraction and solution models are the same?
>
> We want to emphasize that the abstraction and solution generator models are both Qwen 3 1.7B models, exactly the same size and architecture. The only distinguishing difference between them is their initialization, which is the warmstart procedure described above for abstraction generation.

---

> ### Comment · Reviewer_QvJ9 · 2025-11-23
> **Question on response**
>
> Thanks for the response.
>
> The current version of the PDF (lines 186-190) states:
> "Results in Figure 2 show that conditioning a problem solver on abstractions improves
> accuracy when two conditions hold simultaneously: (i) the abstraction is not too short (e.g., not just a
> few words that are not informative; example in Appendix D.2) and is generated by a strong generator
> (o4-mini) and (ii) the solution generator has sufficient instruction-following capability (Qwen3-1.7B
> or Qwen3-4B) of interpreting and utilizing the generated abstraction. "
>
> Please advise where the paper says the results hold when the same model is used for abstractions and solutions across the board.

---

> > ### Author Response · Authors · 2025-11-23
> > **Official Comment by Authors**
> >
> > Thank you for responding to us so promptly! We realize the current draft may blur two separate types of experiments in the paper, giving rise to this confusion. Below we clarify this precisely.
> >
> > 1. Section 4 is only a motivating section for evaluating whether abstractions exist and whether conditioning on them is beneficial. To answer this conceptual question, we use a stronger model (o4-mini) to generate candidate abstractions and evaluate them using weaker models of different sizes. This section is purely diagnostic and is not indicative of results of RLAD, our approach.
> >
> > 2. for our method, we always train and evaluate the abstraction generator and solution generator using the same base model within each domain. In more detail, here is an explicit list of all experiments that use the same model for both abstraction generation and solution generation:
> >
> >      - **Math reasoning experiments**: As shown in Section 5.2, we warm-start the abstraction generator by running SFT for 5 epochs on Qwen3-1.7B, and the solution generator is the same Qwen3-1.7B model without warm-start SFT. This ensures matched model capacity throughout all math experiments. Also as shown in Table 2, we found that the effective performance of RLAD is largely because of the RLAD approach and not this SFT.
> >
> >      - **ARC-AGI experiments**: In the ARC-AGI program synthesis experiments (Section 4.2 and Appendix B.3), both the abstraction generator and the solution generator are Qwen3-4B models. ARC-AGI is a substantially more complex multimodal/structural reasoning domain than math, and we follow prior work by using Qwen3-4B (as also reflected in Table 1). The warm-start for ARC uses human-augmented BARC abstractions, but the base model architecture for both components is consistently Qwen3-4B.
> >
> > We have revised the paper to make this explicit in both Section 5.2 and Appendix B.3. Let us know if anything is still remaining unclear.

---

> ### Comment · Reviewer_QvJ9 · 2025-11-23
> **and...**
>
> Thanks for the response.
>
> Section 5.2  (lines 316-321) now says:
>
> Practical approach and algorithm details. For warmstarting the abstraction generator, we utilize
> abstractions generated by o4-mini. We then use a weaker model (GPT-4.1-mini) to evaluate each
> abstraction by comparing solution success with and without it, retaining only those that improve
> performance to form our seed set. Then, we run SFT of Qwen3-1.7B for 5 epochs on this seed dataset
> to obtain our initial abstraction generator (Qwen Team, 2025). For the solution generator, we use the
> same Qwen3-1.7B model, ensuring that both components use models of identical capacity.
>
> Please advise which results in the paper are a level comparison where warmstart, training and inference are all on the same model.   Thanks

---

> > ### Author Response · Authors · 2025-11-23
> > **Official Comment by Authors**
> >
> > As clarified in the updated Section 5.2, the stronger models (o4-mini for abstraction generation and GPT-4.1-mini for filtering) are used once, prior to any training, solely to construct an initial seed set of candidate abstractions. **They are never used during RLAD training, nor do they ever serve as the abstraction generator or solution generator**. Therefore, **all results in Table 2 and throughout Section 6 reflect a true level comparison on the same Qwen3-1.7B model**. Specifically, Qwen3-1.7B is used for (1) warm-starting the abstraction generator via SFT on the filtered set, (2) RLAD training for both components, and (3) all inference-time evaluations (w/o abstractions, w/ abs (avg), and w/ abs (best)). The reason why we need to use warmstart is to seed the notion of an abstraction in these reasoning models.
> >
> > We also want to highlight that **warm-start alone does not yield performance improvements**. Table 2 shows that conditioning Qwen3-1.7B on SFT-only abstractions ("w/ abs (avg)" in the SFT row) does not consistently outperform the "w/o abs" baseline, and in some settings, e.g. AMC 2023, performs worse. This indicates that naïvely distilling abstractions does not replace RLAD. These observations highlight the necessity of the full RLAD training procedure: only after RL optimization do abstractions become systematically useful, and only then does abstraction-conditioned inference produce the consistent gains reported across benchmarks. And **RLAD uses the same models throughout Section 6**.

---

> > > ### Comment · Reviewer_QvJ9 · 2025-11-24
> > > **...and...**
> > >
> > > Thanks for the response.
> > > The new material in lines 395-403 strengthens the case that the abstraction generator contributes to performance gains. would help to explicitly specify which model was used in the abstraction generator in that experiment.
> > >
> > > Maintaining my accept rating.

---

> > > > ### Author Response · Authors · 2025-11-24
> > > >
> > > > For the weak-to-strong generalization experiments (lines 395–403), we employ a weaker abstraction generator, the same Qwen 3 1.7B model used in Table 2 (Section 6.1), together with a stronger solution generator (o4-mini). This setup highlights that even for frontier models like o4-mini, incorporating abstractions leads to measurable performance gains. We will revise the manuscript to make this clearer.

---

> > > > > ### Comment · Reviewer_QvJ9 · 2025-11-24
> > > > > **...one more thing...**
> > > > >
> > > > > Thanks for clarifying.
> > > > >
> > > > > Perhaps Section 4 can position the asymmetric warmstart as strength rather than a weakness.   Seems the asymmetry is a form of "Curriculum Training" where a stronger abstraction generator earns easy rewards in aiding a weaker solver...

---

> > > > > > ### Author Response · Authors · 2025-11-26
> > > > > >
> > > > > > Thank you for the suggestion. We will incorporate this discussion into Section 4.

---

### Official Review · Reviewer_zyoJ · 2025-10-31

**Soundness:** 3
**Presentation:** 3
**Contribution:** 3
**Rating:** 8
**Confidence:** 3

**Summary:**

This paper proposes RLAD, a way to train LLMs to actually discover and use reasoning abstractions instead of just writing longer chains of thought. The idea is to have one model generate short natural language “abstractions” (like useful hints or lemmas) and another model solve the problem using them. Both are trained together with RL so the abstraction model gets rewarded when its hints help the solver. It basically teaches the model to explore breadth instead of just depth in reasoning. Across math benchmarks like AIME and DeepScaleR, it beats long-CoT RL methods by a large margin and shows that generating abstractions is a better use of compute than brute-force sampling.

**Strengths:**

The paper presents a novel and well motivated idea which is using reinforcement learning to discover and leverage reasoning abstractions, rather than merely extending reasoning depth which is what typical RL does. The approach is technically sound, with clear definitions, careful reward design, and strong empirical validation. The 2 player design is original and The writing is clear and well-structured, making a fairly complex setup easy to follow. Overall, it stands out as an original and solid contribution that meaningfully advances how we think about structured reasoning in large language models.

**Weaknesses:**

The reward design, while intuitive, could be better analyzed to show why the two-player setup works beyond simple prompt diversity effects. The paper would also benefit from more interpretability analysis of the discovered abstractions how stable they are, whether they transfer across domains, and what kind of procedural knowledge they actually encode. Finally, the training setup remains computationally expensive, which may limit broader adoption without lighter or more scalable variants.

**Questions:**

How sensitive is RLAD to the relative strength of the abstraction generator versus the solution generator? It seems like training stability could depend heavily on their initialization or model size. could the authors clarify if balancing their capabilities was necessary for convergence or performance?

Can the authors provide a deeper analysis of what kinds of abstractions emerge during training, for example, are they procedural (methods), factual (lemmas), or meta-level strategies and whether these abstractions transfer effectively to tasks outside math and ARC-AGI?

---

> ### Author Response · Authors · 2025-11-23
> **Official Comment by Authors**
>
> Thank you for the insightful questions! We have added several analyses and clarifications: (1) a characterization of emergent abstractions during RLAD, showing they include procedural, factual, and meta-level strategies; (2) clarification that RLAD’s training cost is similar to standard multi-turn GRPO and easily implemented in existing RL frameworks; (3 analysis explaining why abstractions provide gains beyond simple sampling or prompt-diversity effects; and (4) results showing RLAD is robust even when the abstraction generator is much weaker than the solver.
> **Please let us know if you find your concerns addressed, and if so we would be grateful if you are willing to accept the paper.**
>
> > Can the authors provide a deeper analysis of what kinds of abstractions emerge during training, for example, are they procedural (methods), factual (lemmas), or meta-level strategies and whether these abstractions transfer effectively to tasks outside math and ARC-AGI?
>
> This is a great question! We ran a new study to characterize the nature of abstractions that emerge after training, analyzing the model’s generated abstractions on the held-out set of AIME 2025.  Qualitatively, abstractions are a combination of procedural, algorithmic, and factual knowledge as you describe. Below, we use an LLM as a judge (using GPT 5.1) to classify abstractions from AIME 2025 into these three categories, allowing the judge to select multiple options if they apply (resulting in a non 100% sum):
>
> | Classification Type   | Normalized Count |
> |------------------------|------------------|
> | Procedural             | **67%**          |
> | Meta-level strategy    | **70%**          |
> | Factual                | **53%**          |
>
>
> Note, we impose no constraint on the form of abstraction that is generated by the model. We replicate the efficacy of abstraction generation in three distinct domains: (1) Math Reasoning, (2) ARC-AGI, and (3) Text Classification (with 47 subdomains), showcasing the efficacy of the algorithm across domains and that this approach is not narrow.
>
> > Training setup remains computationally expensive, which may limit broader adoption without lighter or more scalable variants.
>
> Note that training the model should not be any more computationally expensive than fine-tuning a multi-turn model, with two turns, with GRPO with a larger number of samples per prompt. The only difference is the aggregation of rewards per turn. Several recently released open-source frameworks, such as SkyRL[1], VeRL[2], PipelineRL[3], and TRL[4] can be used to readily implement this approach.
>
> > The reward design, while intuitive, could be better analyzed to show why the two-player setup works beyond simple prompt diversity effects.
>
> This is a great question! The purpose of the iso-compute experiments was to see how much of the contribution of abstractions can be obtained via a simple pass@k sampling approach, which is a really effective way of promoting sample diversity. As you can see in our results in Figure ??, it is better to instead use more sampling budget on producing abstractions, meaning that using abstractions indeed does better than simply scaling up sampling.
>
> Of course, we have not yet analyzed the diversity obtained via different system instructions – and will aim to do that next – but we believe that solution generators like Qwen3-1.7B that are trained entirely for reasoning are expected to illustrate minimal sensitivity to system instruction or minor changes in the prompt. An abstraction enriches the prompt with additional information that is indeed referrable and useful, going beyond rephrasings or small tweaks to the system instruction or prompt. We therefore hypothesize that using abstractions produced by the abstraction generator would perform better than simple ways to improve diversity.
>
> > How sensitive is RLAD to the relative strength of the abstraction generator versus the solution generator? It seems like training stability could depend heavily on their initialization or model size. could the authors clarify if balancing their capabilities was necessary for convergence or performance?
>
> In Section 6.2, we showcase the strength of the model for weak-to-strong generalization, where the solution generator is significantly stronger than the abstraction generator, and still show the benefits of the proposed abstraction vs generating a new solution. This indicates that even with a mismatch of model size/capability, generating abstractions is still beneficial.

---

### Official Review · Reviewer_w65f · 2025-11-01

**Soundness:** 3
**Presentation:** 3
**Contribution:** 2
**Rating:** 6
**Confidence:** 4

**Summary:**

The authors propose a two-player RL paradigm where they train a model that proposes "reasoning abstractions" based on the question and another model that actually learns to solve the problem with the proposed abstraction. They show that this leads to meaningful improvements against a baseline RL policy trained to simply solve the questions.

**Strengths:**

- The paper is written clearly and their contribution in terms of the proposed algorithm and its effectiveness over a the DAPO-based RL baseline is both intuitive and clear.
- Show some interesting analyses on how to allocate compute between the abstraction generator and the solver and points out that allocating more compute to the generator is often times more helpful than the solver.

**Weaknesses:**

- It is unclear that current models are not already good enough at generating abstractions based on their solutions. For example, [1] and [2] show that one can prompt LLMs to generate their own abstractions and create a "cheatsheet" of abstractions (whether it's cumulative or using a retriever), which also boosts their performance. So, a comparison to a baseline where one can simply prompt the model to generate its own abstraction would be compelling.
- Some details on the training are unclear: the generator and solver are different models? How many samples were used to run the warm-start of the generator? (how many abstractions per question were generated per problem for the warm start)

[1] Suzgun, Mirac, et al. "Dynamic cheatsheet: Test-time learning with adaptive memory." arXiv preprint arXiv:2504.07952 (2025).
[2] Didolkar, Aniket, et al. "Metacognitive reuse: Turning recurring LLM reasoning into concise behaviors." arXiv preprint arXiv:2509.13237 (2025).

**Questions:**

- How well does the abstraction generator generalize across different datasets? For example, can they be trained on AIME and then transferred to other math datasets? Or does it overfit to that distribution of math problems.
- Also, how much delta/performance gain are we getting from training both the solver and generator? If the solver does not go through the RL training and the only abstraction generator does, does the performance gain vary significantly?

---

> ### Author Response · Authors · 2025-11-23
> **Official Comment by Authors**
>
> Thank you for the thoughtful feedback! To address your concerns, we have added several new results and clarifications: (1) we compare our two-model design with a **joint model** that generates both abstraction and solution in one pass, finding **negligible difference in performance**; (2) we clarify that prior abstraction-style methods rely on **much larger frontier models** and purely inference-time mechanisms, whereas our approach induces abstraction behavior with a **1.5B academic-budget model**; (3) we detail our warm-start procedure (≈4000 OmniMath problems) and show that the abstraction generator generalizes well to fully unseen datasets such as AIME 2025, HMMT, and DeepScaleR-Hard; and (4) we show that RL training and abstractions are complementary, as only their combination yielding the strongest gains. **Please let us know if you find your concerns addressed, and if so we would be grateful if you are willing to accept the paper.**
>
> > It is unclear whether current models are not already good enough at generating abstractions based on their solutions.
>
> It is important to note that prior work, such as Dynamic Cheatsheet [1], relies on a frontier model (Claude 3.5 Sonnet) for memory generation. In that prior work, no training of any form is done, and the authors propose a purely inference-time technique for synthesizing and using memory. Furthermore, MetaCognitive Reuse [2], **which was released after the ICLR 2026 submission deadline**, employs substantially larger models (R1-llama-70B and Qwen3-32B) during inference to generate meta-cognitive behaviors. In contrast, our approach operates within an “academic” budget, utilizing a significantly smaller 1.5B model for both training and inference. Given this parameter constraint, explicit priming is necessary to induce the desired behaviors. Note that priming the model with abstraction can be achieved through human annotations, as seen in the ARC-AGI setup, making it possible without the need for an explicit stronger teacher.
>
> > Clarification: Are the generator and solver different models?
>
> We currently instantiate the generator and solver as separate models. They interface during inference, where the generator's abstraction is provided as input to the solver. During training, the generator is optimized using a reward function based on the solver's success rate. Conversely, the solver is trained on examples both with and without the generated abstraction in its context. In principle, nothing in our method requires these to be separate networks; a single joint model could generate both the abstraction and the final solution.
>
> **New results for joint models**. To address the reviewer’s suggestion, we additionally trained a joint model that generates the abstraction and solution sequentially within a single forward pass. Specifically, we warm-start the model using concatenated abstraction–solution traces, and during training prompt it to first generate an abstraction and then a solution conditioned on that abstraction. However, such a design typically demands substantially larger model capacity, and, more importantly, we find empirically that the performance gap between a joint model and our two-model setup is negligible. We will add this clarification to the appendix.
>
> | Method                 | AIME 2025 Pass@1 w/o abs | AIME 2025 Pass@1 w/ abs |
> | ---------------------- | ------------------------ | ----------------------- |
> | Qwen-3-1.7B            | 33.75                    | 36.25                   |
> | + RLAD (Ours)          | **38.04**                | **42.45**               |
> | + Joint Model         | **37.52**                | **41.33**               |

---

> > ### Author Response · Authors · 2025-11-23
> > **Official Comment by Authors**
> >
> > > Clarification: How many samples were used to run the warm-start of the generator?
> >
> > We used a subset of roughly 10% of the problems from DeepScaleR [1], specifically the problems corresponding to OmniMath [2] with their solutions (approximately 4000 problems) to initialize the abstraction generator. Given that Omnimath has only a single solution per problem, we generate 1 abstraction per problem.
> >
> > > How well does the abstraction generator generalize across different datasets? For example, can they be trained on AIME and then transferred to other math datasets? Or does it overfit to that distribution of math problems?
> >
> > We explicitly see both in distribution and out of distribution generalization of abstractions across different mathematical datasets. To clarify, crucially note that our model is not trained on AIME and it is only used as an evaluation set. We use the identical training mixture as DeepScaleR [1] and e3 [2], which is strictly decontaminated against our evaluation sets (AIME 2025, HMMT, and DeepScaleR-Hard). The results in Table 2 demonstrate robust performance across these unseen datasets, confirming that the generator generalizes well and does not overfit.
> >
> > > If the solver does not go through the RL training and the only abstraction generator does, does the performance gain vary significantly?
> >
> > Table 2 demonstrates that the base model equipped with abstractions (Row 1, Column 2) achieves an accuracy of 36% on AIME 2025. When RL fine-tuning is applied to the solver in conjunction with abstractions (Row 3, Column 2), performance improves significantly to 42%. Conversely, RL fine-tuning without abstractions yields only 38% accuracy (Row 2, Column 1). These results indicate that combining RL training with abstraction generation is essential for maximizing performance.

---

> > > ### Author Response · Authors · 2025-11-26
> > > **Did our Rebuttal Address your Concerns?**
> > >
> > > Dear Reviewer,
> > >
> > > Since there is only 1 week left in the rebuttal period, it would be helpful for us to know if our rebuttal above addressed all your questions and concerns or if any others still remained? We are happy to discuss further and provide more evidence if it is helpful. If all your concerns are addressed we would be grateful if you could acknowledge that soon.
> > >
> > > Thanks,
> > >
> > > Authors

---

### Official Review · Reviewer_JioX · 2025-11-03

**Soundness:** 2
**Presentation:** 3
**Contribution:** 3
**Rating:** 4
**Confidence:** 4

**Summary:**

This paper introduces RLAD, an approach to RL post-training of language models. The approach trains both a reasoning abstraction generator and an abstraction-conditioned solution generator in a cooperative two-player game set-up. Experiments demonstrate the success of the approach, and analyses demonstrate the utility of reasoning abstractions.

**Strengths:**

1. *Originality:* Although the idea of reasoning abstractions has existed for a long time, this is likely the first work that explicitly uses RL to train the generation of reasoning abstractions as part of RL post-training.
2. *Quality:* Comprehensive experiments and analyses are used to demonstrate the utility of reasoning abstractions for solution generation and training to generate them.

**Weaknesses:**

### A. Evaluation

1. My main concern is with the rigor of evaluation, where there’s a reasonable doubt of unfair comparison. The model was warm-started with SFT data in the beginning, similar to DeepSeekR1, but was compared with DAPO, which doesn’t have warm-starting*.

*I’m not completely sure about this since I couldn’t find details on the DaPO baseline in the paper—also see my Question 3 below.

2. Furthermore, the SFT data for the abstraction generator was generated with models (o4-mini) significantly stronger than the models being post-trained (Qwen3-1.7B), which raises the question of how well the approach scales/how RLAD would be applied to models that are already among the strongest.

3. The ablation studies (Appendix B2) do not study how much the performance gains are from the SFT data generated with stronger models vs. other parts of the training pipeline. (Also see Question 4.)

4. The definition of iso-compute scaling curves is not well-motivated.
  - It is unclear what $k_0$ represents in the formula $m \times (k - k_0) = \mathcal C$, and why this formula is supposed to capture the notion of “iso-compute”. Also, although there is a high-level explanation about how the offset discounts samples that are too similar, subtracting an offset appears to be an arbitrary choice relative to many other possible options such as dividing by a constant.
  - Even in the formula $m \times k = \mathcal C$, this does not capture the extra compute used to generate the abstractions.

### B. Related Work section

There is, in my opinion, an important line of past work that has been omitted in the Related Work section, namely, past work on learning and using reasoning abstractions. There’s a significant body of literature on learning and using abstractions (also often called concept learning or library learning) – see, e.g., [1], [2], [3], [4].

DreamCoder, LaSR, LINC, LEMMA,

[1] Ellis, Kevin, et al. "Dreamcoder: Bootstrapping inductive program synthesis with wake-sleep library learning." *PLDI*, 2021.

[2] Li, Zhening, et al. "LEMMA: Bootstrapping High-Level Mathematical Reasoning with Learned Symbolic Abstractions." *MATH-AI Workshop at NeurIPS’22*, 2022.

[3] Grand, Gabriel, et al. "LILO: Learning Interpretable Libraries by Compressing and Documenting Code." *ICRL*, 2024.

[4] Grayeli, Arya, et al. "Symbolic regression with a learned concept library." *NeurIPS*, 2024.

### C. Presentation

1. The main number claimed in the introduction is “44% improvement over… DAPO on AIME 2025”, which should be clarified to mean relative (percentage) improvement instead of absolute improvement. Furthermore, 44% appears inconsistent with the reported numbers in the tables. In Table 2, the RLAD w/ abs (avg) result on AIME 2025 42.45%, and the DaPO w/o abs (avg) result is 37.92%, and 42.45/37.92 = 1.119, which is far from the claimed 1.44. (I did manage to find a ratio that’s close to 1.44, aka the ratio of RLAD w/ abs (best) 48.33% to no RL post-training w/o abs (avg) 33.75% is 1.432, but a) these are not the right numbers to take the ratio of, and b) 1.432 should round to 1.43 and not to 1.44.)

2. The way the paper is presented suggests that “reasoning abstractions” is a novel concept introduced by the paper, which is far from reality (see Weakness B on Related Work). The novel aspect of the paper is, in my opinion, not the idea of “reasoning abstractions” themselves, but the use of RL to post-train a model to generate reasoning abstractions in tandem with an abstraction-conditioned solution generator (last three lines of the Related Work section). Clarification in the paper would be appreciated.

3. High-level presentation would benefit from more explicit summaries at the beginning of each (sub)section. For example, the summary paragraph at the beginning of Section 4 would benefit from a sentence like “In this section, we describe our approach to generating reasoning abstractions via summarization of reasoning traces, and demonstrate how they improve reasoning performance by acting as hints.”

4. The “Modifying reward design” paragraph is not super intuitive and would benefit from further elaboration on how it “address[es] one of the challenges” and which challenge it addresses. Currently, the reward design appears to be mainly supported by the empirical fact that it improved performance (Appendix B2), but motivation from first principles would be nice to see.

5. Typos:
- Incorrect citations, e.g., DeepScaleR Hard (line 81-82) mistakenly cited e3.
- Line 116: “(a) … (c) …” -> “(a) … (b) …”
- Lines 260-261: “recent results RL”
- Inconsistent capitalization of DAPO as DAPO in the main text and DaPO in the appendices.

6. Appendix C is empty.

**Questions:**

1. For ARC, I’m curious how the results would compare with another baseline: best-of-N. It is known that in domains where external verification is available (such as ARC), best-of-N is a powerful strategy. So I’m wondering, is adding abstraction generation more cost-effective or best-of-N more cost-effective (Table 1)?

2. Lines 70-71 says “The abstraction generator is rewarded for the improvement in the
accuracy of the solution generator, stemming from conditioning on the abstractions it proposes.”
But Equation 2 is the expected reward of the solver conditioned on z, whereas from that sentence I would’ve expected it to be difference w.r.t. the reward of the solver without z. So does the actual reward subtract or not subtract the solver’s reward without conditioning?

3. Did training the DAPO baseline include SFT warm-starting similar to that used in RLAD? (Weakness A1 and Question 4 currently assume the answer is “no”, but I’m not completely sure because I didn’t find any details regarding the DAPO baseline in the paper.)

4. What is the performance of RLAD without warm-starting? Alternatively, how does RLAD compare with a version of DAPO that includes a similar warm-starting procedure? (Also see Weakness A1 regarding fair comparison.)

---

> ### Author Response · Authors · 2025-11-23
> **Official Comment by Authors**
>
> Thank you for the feedback! To address your concerns, we have now added several new experimental results: 1) a comparison between RLAD and a warmstarted DAPO baseline, 2) an experiment evaluating the efficacy of RLAD and our abstractions with frontier models, 3) an experiment showing that the gains in RLAD stem from the RLAD method itself and not warmstarting data. All of these experiments clearly reveal that the gains with RLAD stem from the approach itself and not the other factors, and that RLAD is effective at scale as well. We have added these clarifications to the paper and clarified them below. We also provide clarifications on other questions. **Please let us know if you find your concerns addressed, and if so we would be grateful if you are willing to accept the paper.**
>
> > A1. My main concern is with the rigor of evaluation, where there’s a reasonable doubt of unfair comparison. The model was warm-started with SFT data in the beginning, similar to DeepSeekR1, but was compared with DAPO, which doesn’t have warm-starting*.
>
> We would like to clarify that **only the abstraction generator was warm-started**. **The solution generator, the model ultimately compared under DAPO vs. RLAD, is not warm-started with any additional data**. It is trained from the same base model (Qwen3-1.7B) using either DAPO or RLAD. The prompts are identical across methods; the only difference is that RLAD has access to the abstractions produced by the abstraction generator for training.
>
> That said,  to address your concern fully, we have now added an updated version of the DAPO baseline that more closely mirrors the RLAD setup. Specifically, we first SFT the base model for DAPO on the concatenation of abstraction + solution, and then train this SFT-warm-started model with DAPO on the exact same dataset used for RLAD. On AIME2025, this new baseline achieves:
>
> | Method                 | AIME 2025 Pass@1 w/o abs | AIME 2025 Pass@1 w/ abs |
> | ---------------------- | ------------------------ | ----------------------- |
> | Qwen-3-1.7B            | 33.75                    | 36.25                   |
> | + DAPO                 | 37.92                    | 34.90                   |
> | + DAPO (w/ warm-start) | 35.33                    | 34.30                   |
> | + RLAD (Ours)          | **38.04**                | **42.45**               |
>
> We observe that the performance, particularly in the “with abstraction” variant, is only marginally better than the base model and similar to the original DAPO baseline We also observed that the entropy of the token distributions induced by the warmstarted model was substantially lower than the Qwen3-1.7B base model used as the solution generator by RLAD, which is perhaps expected and a drawback of SFT in general.
>
> Beyond changes in entropy, our analysis also indicates that this happens because the model tends to unlearn abstraction generation during DAPO training: even after SFT, due to stochastic sampling, the model frequently produces solutions without abstractions. These trajectories can still receive positive reward under DAPO, which reinforces the behavior of skipping abstractions, causing the model to drift away from using them. In contrast, RLAD explicitly enforces both (1) accurate solution generation and (2) maintenance of abstraction-generation capability. This prevents the “unlearning” effect and enables the model to explore a richer solution space throughout training. **As a result, RLAD consistently outperforms all DAPO variants.**

---

> ### Author Response · Authors · 2025-11-23
> **Official Comment by Authors**
>
> > A2. Furthermore, the SFT data for the abstraction generator was generated with models (o4-mini) significantly stronger than the models being post-trained (Qwen3-1.7B), which raises the question of how well the approach scales/how RLAD would be applied to models that are already among the strongest.
>
> We would like to clarify that our method does **not** rely on o4-mini specifically. The only requirement for the abstraction generator is the ability to perform summarization or produce concise abstractions given a problem. Any competent summarizer works; we chose o4-mini to demonstrate that this doesn’t constrain scaling. As shown in Figure 2 (left) in the submission, **the solution generator, not the abstraction generator, is usually the bottleneck**, where the solution generator is unable to follow the proposed abstraction when weaker. RLAD directly improves this solution-generation capability without requiring expert supervision.
>
> To further address the reviewer’s question about applicability to stronger models, we **include an experiment (Section 6.1)** demonstrating that our pipeline remains beneficial even when the solution generator is a frontier model and is even stronger compared to the abstraction generator. We pair our trained abstraction generator with o4-mini as the solution generator and evaluate under a fixed 24k-token budget with 4 samples per question. Without abstractions, o4-mini achieves 80.38% pass@1, 82.26% pass@2, and 84.77% pass@4. Conditioning on both the problem and the proposed abstractions improves performance to 85.83% (+5.45%) pass@1, 88.33% (+6.07%) pass@2, and 90.00% (+5.23%) pass@4 accuracy. Even o4-mini, the model used to generate our SFT data, **gains >5% pass@1 when conditioned on abstractions from our trained generator**. This demonstrates that RLAD’s abstraction-conditioning mechanism scales effectively and provides benefits even when applied to already-strong models.
>
> > A3. The ablation studies (Appendix B2) do not study how much the performance gains are from the SFT data generated with stronger models vs. other parts of the training pipeline. (Also see Question 4.)
> We would like to clarify that the SFT stage is applied **only to the abstraction generator**, not to the solution generator. Thus, the solution-generation capability of the final model does not directly benefit from any strong-model SFT beyond the abstractions themselves.
>
> To isolate how much of the performance gain comes specifically from the SFT-generated abstractions versus the rest of the RLAD pipeline, the most relevant comparison is within Table 2. On AIME2025:
> | Configuration          | Accuracy        |
> | ---------------------- | --------------- |
> | Base Qwen3-1.7B (no abstraction)  | 33.75%          |
> | + Abstraction SFT only | 36.25% (+2.5pp) |
> | + RLAD (full)          | 42.45% (+8.7pp) |
>
> This shows that providing abstraction generated by an abstraction generator with SFT data alone yields a +2.5% improvement over the base model on AIME2025. However, applying RLAD on top of the same abstraction data produces a much larger gain of +8.7% over the base model (or +6.2% over the SFT-only variant).
>
> Therefore, while the SFT data from stronger models provides a modest benefit, the majority of the improvement comes from the RLAD training itself rather than from warm-starting. Training via RLAD does not use any stronger model in any way, indicating that these gains stem from the efficacy of the method. RLAD leverages abstractions during training to enhance exploration and solution quality far beyond what SFT alone achieves.

---

> > ### Author Response · Authors · 2025-11-23
> > **Official Comment by Authors**
> >
> > > A4.1. The definition of iso-compute scaling curves is not well-motivated. It is unclear what is represented in the formula, and why this formula is supposed to capture the notion of “iso-compute”.
> >
> > Our iso-compute formulation follows established methodology from Snell et al., ICLR 2025 [1], as illustrated in Figure 7 (left). In both studies, a fixed sample budget (depicted in the color bar) is maintained, while the x-axis reflects the varying proportion of samples drawn from one distribution to another.
> >
> > In our formulation, we measure the ratio of newly generated abstractions to new solutions per abstraction, while keeping the total number of samples (abstractions * solutions) constant. Each line in the figure corresponds to a fixed compute budget, and the plotted variation captures how abstractions and solutions trade off under this constraint. This approach provides a consistent way to examine scaling behavior at constant total inference compute."
> >
> > There are other variations possible in this definition, such as measuring the notion of iso-compute by holding the total number of tokens constant, but please note that using this definition will only favor scaling the number of abstractions more than scaling the number of solutions, as abstractions are far more concise than typical reasoning traces conditioned on this abstraction.
> >
> > > A4.2. Also, although there is a high-level explanation about how the offset discounts samples that are too similar, subtracting an offset appears to be an arbitrary choice relative to many other possible options such as dividing by a constant.
> >
> > We acknowledge that several approaches could be used to construct an offset, including dividing by a constant, as the reviewer suggests. However, in our formulation, $N_0 = 0$ (no offset) already clearly illustrates the trend we aim to highlight: with higher compute budgets, generating abstractions becomes more advantageous than generating solutions. The inclusion of additional offset values reinforces this trend, effectively treating $N_0$ samples as “provided for free” to the user, thereby better emphasizing the relationships shown in the plot. However, if the reviewer has a strong objection to this presentation, we are happy to remove the offsets if you think that'll be better. Please let us know!
> >
> > > A4.3. Even in the formula, this does not capture the extra compute used to generate the abstractions.
> >
> > We decided to omit the additional computation required to generate an extra abstraction, since the number of tokens involved is negligible compared to those used to produce a full solution. On average, an abstraction comprises only about 256-512 tokens—insignificant relative to the 16K–32K token budget typically allocated for solution generation—resulting in a negligible impact on iso-compute performance. Moreover, solutions that include an abstraction tend to be shorter than those without one, further reducing any effective cost. We will include a footnote to clarify this point.
> > References:
> >
> > [1] Snell et al. 2024, Scaling LLM Test-Time Compute Optimally can be More Effective than Scaling Model Parameters
> >
> > > B. Related Work section
> > > C.2 Clarifying Contribution of the work
> >
> > Thank you for the valuable feedback. We agree that the general concept of “reasoning abstractions” has been explored in prior work. In the revised version, we will add explicit citations to clarify this context. Our contribution lies in the formulation and training of natural language abstractions: concise, model-generated textual summaries of procedural and factual knowledge rather than symbolic or hand-engineered abstractions used in previous work. Moreover, our method jointly optimizes both the discovery and utilization of these abstractions through a two-player RL framework, where an abstraction generator and an abstraction-conditioned solution generator co-evolve. Furthermore, we showcase that these natural language abstractions are beneficial for exploration, showcasing benefits in the pass@k accuracy as seen in the iso-compute plots presented. While we do not claim to introduce abstractions as a new concept, we believe that our approach of applying reinforcement learning to induce and leverage natural language reasoning abstractions in tandem represents a novel contribution.
> >
> > > C.1 Improvement over DAPO.
> >
> > Thank you for raising this. We will fix this in the revised manuscript.

---

> > > ### Author Response · Authors · 2025-11-26
> > > **Did our Rebuttal Address your Concerns?**
> > >
> > > Dear Reviewer,
> > >
> > > Since there is only 1 week left in the rebuttal period, it would be helpful for us to know if our rebuttal above addressed all your questions and concerns or if any others still remained? We are happy to discuss further and provide more evidence if it is helpful. If all your concerns are addressed we would be grateful if you could acknowledge that soon.
> > >
> > > Thanks,
> > >
> > > Authors

---

### Author Response · Authors · 2025-12-02
**Rebuttal-Period Summary – Key Clarifications & New Results**

We thank the reviewers and the AC for their thoughtful and constructive feedback during the rebuttal period. The discussion has substantially improved the paper’s clarity, empirical grounding, and positioning within the literature. In response, we have added several key clarifications and new results that will be incorporated into the final version. For the convenience of the new AC, in light of the recent ICLR policy change, we summarize these updates below:

- **Warmstart Clarification and Additional Baseline**: We clarify that only the abstraction generator was warm-started; the solution generator, the model ultimately compared under DAPO vs. RLAD, is not warm-started with any additional data. Furthermore, we add an updated version of the DAPO baseline, which directly mirrors the RLAD setup. Here, we first perform supervised fine-tuning (SFT) on the base model on the concatenation of abstraction + solution from the warmstart in RLAD, and second, train with the DAPO objective on the same set of prompts as RLAD. RLAD consistently outperforms this strengthened DAPO baseline, indicating that warm-starting alone is not sufficient to explain the performance gains.

- **Strength of Abstraction vs Solution Generator**: We demonstrate that our pipeline remains beneficial even when the solution generator is a frontier model and is even stronger compared to the abstraction generator. We pair our trained abstraction generator with o4-mini as the solution generator and evaluate under a fixed 24k-token budget with 4 samples per question. Without abstractions, o4-mini achieves 80.38% pass@1, 82.26% pass@2, and 84.77% pass@4. Conditioning on both the problem and the proposed abstractions improves performance to 85.83% (+5.45%) pass@1, 88.33% (+6.07%) pass@2, and 90.00% (+5.23%) pass@4 accuracy. This demonstrates that RLAD’s abstraction-conditioning mechanism scales effectively and provides benefits even when applied to already-strong models.

- **Clarifying Contribution of the work**: Though the general concept of reasoning abstractions has been proposed in prior work, our contribution lies in the formulation and training of natural language abstractions: concise, model-generated textual summaries of procedural and factual knowledge rather than symbolic or hand-engineered abstractions used in previous work. We have updated the related work to reflect this.

- **Can Abstractions Generalize Across Datasets?**: We explicitly evaluate the generalization of abstractions to both in-distribution and out-of-distribution mathematical benchmarks. We train on DeepScaleR, which is strictly decontaminated from our evaluation sets, including AIME 2025, HMMT, and DeepScaleR-Hard. RLAD exhibits robust performance on all of these unseen datasets, providing empirical evidence that the learned abstractions transfer beyond the training distribution.

- **Deeper Qualitative Analysis of Abstractions**: We ran a new study to characterize the nature of abstractions that emerge after training, analyzing the model’s generated abstractions on the held-out set of AIME 2025. We find that qualitatively, abstractions are a combination of procedural, algorithmic, and factual knowledge.

Finally, we note that reviewer QvJ9 engaged extensively with the rebuttal and expressed a favorable view of the paper prior to the review process being cut short.

These additions and clarifications directly address the major concerns raised during the review process. The constructive dialogue has significantly strengthened the work, and we respectfully ask that the AC take these updates into account in their final evaluation.

---

### Meta-Review · Area_Chair_QxVq · 2026-01-05

**Summary:**

This paper introduces a new concept called “reasoning abstractions”, which improves the performance of large language models by jointly training an abstraction generator and an abstraction-conditioned solution generator, and demonstrates that this approach is more effective than simply increasing solution sampling. However, the descriptions of the abstraction generator, the answer generator, and the SFT model are not sufficiently clear and may be misleading to readers; the authors are encouraged to further clarify these components in the revised version. In addition, it would be meaningful to further explain what information is contained in the generated abstractions and why this information helps produce correct answers.

**Reviewer Concerns:**

Reviewer w65f, Reviewer zyoJ, and Reviewer QvJ9 gave positive initial ratings, and the authors actively responded to their concerns during the discussion. For Reviewer JioX, the main concerns were the fairness of experimental comparisons, the discussion of related work, and other presentation issues. During the rebuttal, the authors added corresponding experimental comparisons to demonstrate fairness and revised the related work and text, which is expected to address the reviewer’s concerns.

**Reviewer Scores:**

Reviewer JioX’s main concerns focus on the fairness of the evaluation, such as comparisons with the DAPO method, whether SFT is used for warm-starting, and how the strength of the generator affects the results. During the rebuttal, the authors added corresponding experimental comparisons and ablation studies. In addition, in response to the reviewer’s suggestions regarding related work and presentation, the authors revised the related work section and the corresponding text. These efforts largely addressed the reviewer’s concerns, and the reviewer is likely to increase the score.

Reviewer w65f points out that prior work has shown that directly prompting large language models to generate abstractions can already yield strong performance without special training, and suggests that the paper should compare against this approach. During the rebuttal, the authors clarified that the success of such prompting-based methods heavily depends on frontier-scale models, whereas this work targets smaller academic-scale models. The authors also responded to the reviewer’s questions regarding implementation details and hyperparameter settings. The reviewer is expected to maintain a positive score.

Reviewer zyoJ mainly questions why the abstractions generated by the proposed method are effective and what kinds of useful information they contain. The authors provided explanations and responses, and the reviewer is expected to maintain a positive score.

Reviewer QvJ9’s main concern is whether the performance gains stem from using a stronger model as the abstraction generator. The authors clarified this point. In addition, Reviewer QvJ9 engaged in multiple rounds of discussion with the authors on other details and ultimately decided to maintain the initial positive score.

---

### Decision · Program_Chairs · 2026-01-26

Accept (Poster)